# Convergent use of phosphatidic acid for hepatitis C virus and SARS-CoV-2 replication organelle formation

Keisuke Tabata [1,11,12,16], Vibhu Prasad [1,16], David Paul [1,13], Ji-Young Lee[1], Minh-Tu Pham[1], Woan-Ing Twu[1], Christopher J. Neufeldt [1], Mirko Cortese [1], Berati Cerikan[1], Yannick Stahl [1], Sebastian Joecks[1,14], Cong Si Tran[1], Christian Lüchtenborg[2], Philip V'kovski [3,4], Katrin Hörmann[5], André C. Müller[5], Carolin Zitzmann[6,15], Uta Haselmann[1], Jürgen Beneke[7], Lars Kaderali [6], Holger Erfle[7], Volker Thiel [3,4], Volker Lohmann [1], Giulio Superti-Furga [5,8], Britta Brügger [2] & Ralf Bartenschlager [1,9,10 ✉]

Double membrane vesicles (DMVs) serve as replication organelles of plus-strand RNA viruses such as hepatitis C virus (HCV) and SARS-CoV-2. Viral DMVs are morphologically analogous to DMVs formed during autophagy, but lipids driving their biogenesis are largely unknown. Here we show that production of the lipid phosphatidic acid (PA) by acylglycerolphosphate acyltransferase (AGPAT) 1 and 2 in the ER is important for DMV biogenesis in viral replication and autophagy. Using DMVs in HCV-replicating cells as model, we found that AGPATs are recruited to and critically contribute to HCV and SARS-CoV-2 replication and proper DMV formation. An intracellular PA sensor accumulated at viral DMV formation sites, consistent with elevated levels of PA in fractions of purified DMVs analyzed by lipidomics. Apart from AGPATs, PA is generated by alternative pathways and their pharmacological inhibition also impaired HCV and SARS-CoV-2 replication as well as formation of autophagosome-like DMVs. These data identify PA as host cell lipid involved in proper replication organelle formation by HCV and SARS-CoV-2, two phylogenetically disparate viruses causing very different diseases, i.e. chronic liver disease and COVID-19, respectively. Host-targeting therapy aiming at PA synthesis pathways might be suitable to attenuate replication of these viruses.

[1] Department of Infectious Diseases, Molecular Virology, Heidelberg University, Heidelberg, Germany. [2] Biochemistry Center Heidelberg, Heidelberg University, Heidelberg, Germany. [3] Institute of Virology and Immunology IVI, Bern, Switzerland. [4] Department of Infectious Diseases and Pathobiology, Vetsuisse Faculty, University of Bern, Bern, Switzerland. [5] CeMM Research Center for Molecular Medicine of the Austrian Academy of Sciences, Vienna, Austria. [6] Institute of Bioinformatics and Center for Functional Genomics of Microbes, University Medicine Greifswald, Greifswald, Germany. [7] BioQuant, Heidelberg University, Heidelberg, Germany. [8] Center for Physiology and Pharmacology, Medical University of Vienna, Vienna, Austria. [9] Division Virus-Associated Carcinogenesis, German Cancer Research Center, Heidelberg, Germany. [10] German Center for Infection Research, Heidelberg Partner Site, Heidelberg, Germany. [11] Present address: Department of Genetics, Graduate School of Medicine, Osaka University, Osaka, Japan. [12] Present address: Laboratory of Intracellular Membrane Dynamics, Graduate School of Frontier Biosciences, Osaka University, Osaka, Japan. [13] Present address: MRC Laboratory of Molecular Biology, Francis Crick Avenue, Cambridge CB2 0QH, UK. [14] Present address: LI-COR Biosciences GmbH, Siemensstrasse 25A, Bad Homburg, Germany. [15] Present address: Los Alamos National Laboratory, Theoretical Biology and Biophysics, Los Alamos, NM, USA. [16] These authors contributed equally: Keisuke Tabata, Vibhu Prasad. ✉email: ralf.bartenschlager@med.uni-heidelberg.de

Chronic hepatitis C and COVID-19 are major medical problems. Both diseases are caused by viral infections inflicting a large number of people and having led to millions of deaths[1,2]. Chronic hepatitis C is caused by persistent infection with the hepatitis C virus (HCV), while COVID-19 is due to acute infection with the severe acute respiratory syndrome coronavirus-2 (SARS-CoV-2). Both viruses are biologically very distinct e.g. by having a very narrow tropism and a predominantly persistent course of infection in the case of HCV, contrasting the rather broad tropism and acute self-limiting course of infection in the case of SARS-CoV-2. This biological distinction is reflected by their phylogenetic distance with HCV belonging to the *Flaviviridae* and SARS-CoV-2 being a member of the *Coronaviridae* virus family[3]. In spite of these differences, both viruses possess a single strand RNA genome of positive polarity that is replicated in membranous vesicles in the cytoplasm of infected cells[4,5]. These vesicles are induced by viral proteins, in concert with cellular factors, and composed of two membrane bilayers, thus corresponding to double-membrane vesicles (DMVs). These DMVs accumulate in infected cells and can be regarded as viral replication organelle. Viral DMVs have morphological similarity to autophagosomes[6,7], but while autophagy-induced DMVs serve to engulf cellular content and damaged organelles for subsequent degradation, viral DMVs create a conducive and protective environment for productive viral RNA synthesis. For HCV and SARS-CoV-2, DMVs are derived from the ER[8–10] and can be induced by the nonstructural proteins (NS)3, 4A, 4B, 5A and 5B in the case of HCV[7] and the viral proteins nsp3-4 in the case of MERS-CoV and SARS-CoV[11,12], alongside with co-opted host cell proteins and lipids. Here, we show that common host cell factors are exploited by the phylogenetically distant HCV and SARS-CoV-2 to build up their cytoplasmic replication organelle.

## Results

**Identification of AGPATs as critical host cell factors contributing to HCV replication.** Using HCV as model to study DMV biogenesis, we purified DMVs under native conditions and determined their molecular composition by proteomic profiling (Fig. 1a, b and Supplementary Fig. 1a). To this end we employed human hepatoma cells (Huh7) containing a self-replicating HCV replicon RNA (designated sg4B[HA]31R[13]) in which NS4B was HA-tagged. This RNA replicates autonomously and induces an extensive array of DMVs that can be isolated by HA-affinity purification[13]. Mass spectrometry-based proteomics analysis identified a total of 1487 proteins significantly enriched in the NS4B-HA sample relative to the untagged technical negative control (using SAINT average *P*-values > 0.95) (Supplementary Data 1). Label free quantitation (LFQ) revealed a major overlap of proteins (1542) between the NS4B-HA complex and HCV-naive ER membranes purified in parallel from Huh7 cells stably expressing HA-tagged Calnexin (CNX-HA) (Fig. 1b and Supplementary Fig. 1b). Of note, 309 proteins were significantly enriched in the NS4B-HA sample relative to this ER control with an over-representation of proteins involved in RNA metabolism, intracellular vesicle organization and transport as well as endomembrane organization (Supplementary Fig. 2).

Given our interest in identifying proteins of relevance for DMV formation, we selected 139 candidates with a bias for proteins involved in vesicle transport and biogenesis as well as lipid metabolism. These candidates were validated with respect to their role in HCV replication by using RNA interference-based screening (Fig. 1c and Supplementary Data 2). In this way, we could validate 38 hits as HCV dependency factors. Amongst identified hits were acylglycerolphosphate acyltransferase (AGPAT) 1 and 2, two enzymes that catalyze the de novo formation of phosphatidic acid (PA), a precursor to di- and triacylglycerols as well as all glycerophospholipids[14,15]. In addition, PA is involved in signaling and protein recruitment to membranes and, owing to its small and highly charged head group, promotes membrane curvature[16–18]. Since these properties might be involved in DMV formation, we focused our subsequent analysis on AGPATs.

AGPATs play crucial roles in lipid homeostasis, because enzyme-inactivating mutations in AGPAT2 are linked to congenital generalized lipodystrophy and defects in PA metabolism as well as autophagy are associated with neurological disorders and chronic obstructive pulmonary disease[18,19]. Moreover, severe lipodystrophy as well as extreme insulin resistance and hepatic steatosis have been observed in AGPAT2$^{-/-}$ mice[14]. To date, 11 AGPATs have been identified in mammalian cells. AGPAT1 to 5 preferentially utilize lysophosphatidic acid (LPA) as an acyl donor while AGPAT6 to 11 preferentially utilize alternative lysophospholipid substrates or have a preference for glycerol-3-phosphate. Thus, only AGPAT1 to 5 functions as true LPA acyltransferases[14]. To establish which AGPAT family members are found in NS4B-associated membranes, FLAG-tagged versions of each of the 5 AGPATs were transiently expressed in cells containing the HCV replicon sg4B[HA]31R (Supplementary Fig. 3a). Pull-down of NS4B-HA revealed association with AGPAT1 and 2, and to a lesser extent with AGPAT3, but not with AGPAT4 and 5. In addition, endogenous AGPAT1 and 2 were detected in NS4B-HA containing membranes isolated from replicon-containing cells (Fig. 1d), whereas AGPAT 3 was not enriched. Moreover, in HCV-infected cells AGPAT1 and 2 were recruited to NS4B-containing sites that most likely correspond to sites of DMV accumulation[13] (Fig. 1e).

To validate the role of AGPAT1 and 2 in HCV replication, we created knock-out (KO) cell pools using CRISPR/Cas9. Although we observed reduced cell growth of stable double knock-out (DKO) cells 8 days after transduction of guide RNAs, single KO cell pools showed no such decrease in cell growth and could be used for transient knock-out of the other AGPAT gene without impacting cell viability for up to 8 days after transduction (Supplementary Fig. 3b). Using this approach, we observed that AGPAT1/2 DKO impaired lipid droplet formation (Supplementary Fig. 3c–e) as shown previously[20,21], confirming disruption of AGPAT1/2 function. To monitor the impact of single KO and AGPAT1/2 DKO on HCV replication, cells were infected with an HCV reporter virus and viral replication was determined by using luciferase assay. While single KO suppressed HCV replication by ~50–70%, a reduction by ~90% was observed in DKO cells (Fig. 2a). Even stronger replication suppression was observed with a subgenomic replicon (Supplementary Fig. 4a), confirming that AGPAT depletion affected viral RNA replication and not virus entry or assembly. Of note, replication was completely restored by stable expression of AGPAT1 and 2 in DKO cells, which was not the case with either or both enzymatically inactive mutants (Fig. 2b). Restoration of replication by addition of exogenous PA was not successful due to low solubility of this lipid in aqueous media and cytotoxicity exerted by organic solvents.

In the case of Dengue virus (DENV) and Zika virus (ZIKV) also belonging to the *Flaviviridae* family, but inducing morphologically different membrane alterations, i.e., ER membrane invaginations[4], replication was not affected by AGPAT depletion as determined by plaque assay or with a reporter virus (Fig. 2c and Supplementary Fig. 4b, respectively). These results suggest that enzymatically active AGPAT1 and 2 are required for HCV replication with both AGPATs having partially redundant functions.

**Important role of AGPATs in HCV-induced DMVs and accumulation of PA at these sites.** Next, we determined the impact of AGPAT KO on HCV-induced DMV formation. Since

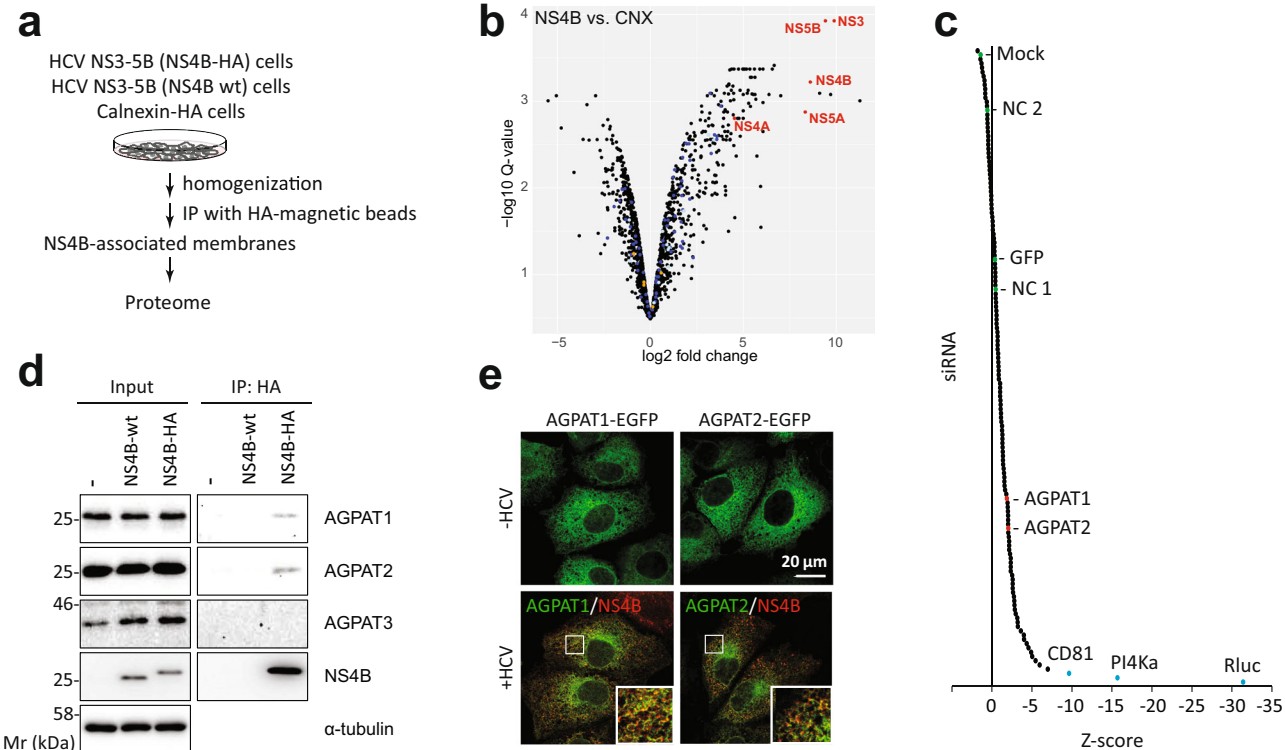

**Fig. 1 Proteome analysis of HCV-induced DMVs identifies AGPATs as host dependency factors critically contributing to viral replication. a**
Experimental approach used to purify DMVs from HCV-replicating cells. **b** Volcano plot of differentially enriched interactors of NS4B and calnexin (CNX).
Q-values were calculated using the limma software package and corrected for multiple hypothesis testing. Viral proteins are highlighted with red letters. A
magnified view with protein hits labeled is given in Supplementary Fig. 1B. **c** A total of 139 genes were selected from the DMV proteome and validated by
siRNA screening (3 siRNAs per gene). CD81, PI4KA and Rluc were used as positive controls; NC (negative control)1, NC2, GFP and mock infection served as
negative controls. A summary of the screening is given in Supplementary Data 2. **d** Endogenous AGPAT1 and 2, but not AGPAT3 are contained in NS4B-
associated membranes. Membranes were purified from naive Huh7-Lunet cells (-), or Huh7-Lunet cells containing a subgenomic replicon without or with an
HA-tag in NS4B (NS4B-wt and NS4B-HA, respectively). Captured proteins were analyzed by western blot, along with the input (2%). α-tubulin served as
loading control. Two biologically independent experiments showed similar results. **e** Colocalization of NS4B with AGPAT1 and 2. Huh7-Lunet cells stably
expressing AGPAT1- or AGPAT2-EGFP were transfected with in vitro transcripts of the HCV genome Jc1 and fixed 48 h post-transfection. Two biologically
independent experiments showed similar results. Source data for panels **c** and **d** are provided as Source Data file.

AGPAT1/2 DKO reduces RNA replication, we employed a
replication-independent system in which DMV production is
induced by the sole expression of an HCV NS3-5B polyprotein
fragment that undergoes self-cleavage to produce functional NS3,
4A, 4B, 5A, and 5B[8,22] (Fig. 3a). To allow detection of the
replicase subcellular location by fluorescence microscopy, NS5A
was fluorescently tagged with EGFP. This tagging has no effect on
replicase functionality[8,22]. While expression of this polyprotein
induced a high number of DMVs in control cells, DMV abun-
dance was dramatically reduced in AGPAT1/2 DKO cells (Fig. 3a,
b), although amounts of viral proteins were comparable in control
and DKO cell pools (Fig. 3c). Moreover, DMVs had a smaller
diameter in AGPAT2 KO cells (Supplementary Fig. 4c). These
results argue for a pivotal role of AGPATs in HCV DMV
biogenesis.

Given that AGPAT1 and 2 are important for DMV formation
and their enzymatic activity is required for HCV replication, we
next focused on their reaction product, i.e. the lipid PA. To
quantify the amount of PA associated with HCV-induced DMVs
and compare it to PA associated with ER membranes, we
determined the lipidome of highly purified DMVs isolated from
cells containing the sg4B$^{HA}$31R replicon (Fig. 3d). Consistent
with earlier results, these membranes contained elevated amounts
of cholesterol and sphingolipids, which served as positive
controls, relative to ER membranes purified in parallel from
Huh7 cells stably expressing HA-tagged Calnexin[13,23]. Of note,

PA abundance in DMVs also was increased in comparison to ER
membranes, whereas the level of diacyl phosphatidylcholine
(aPC) and several other lipids was not affected (Fig. 4d; for
further lipids see Supplementary Data 3).

To confirm these findings at the single-cell level, we used two
alternative methods to detect PA by fluorescence microscopy.
First, we generated a recombinant protein composed of GST that
was fused to the PA-binding domain (PABD) derived from yeast
Spo20p (Supplementary Fig. 5a, b). As specificity control, we
employed the analogous sensor protein containing a mutation in
the PABD that abolishes PA binding, and GST alone[24]. These
proteins were introduced via transient permeabilization into
Huh7 derived cells (Supplementary Fig. 5c). In cells treated with
phorbol 12-myristate 13-acetate (PMA), a potent activator of
phospholipase D-mediated PA production, as expected the intact
sensor predominantly stained the plasma membrane, which was
not the case with the PA non-binding mutant or GST alone,
confirming specificity of the signal (Supplementary Fig. 5d).
Moreover, also in cells that were not treated with PMA, the PA
sensor predominantly stained the plasma membrane (Supple-
mentary Fig. 5d, right panel). Using this assay, we monitored
intracellular PA distribution in HCV replicon-containing cells
and observed PA colocalization with NS4B (Supplementary
Fig. 5e). As second assay for intracellular PA detection, we
created an EGFP-tagged sensor fused to the PABD of Raf1, a
serine-threonine kinase recruited to cellular membranes via its

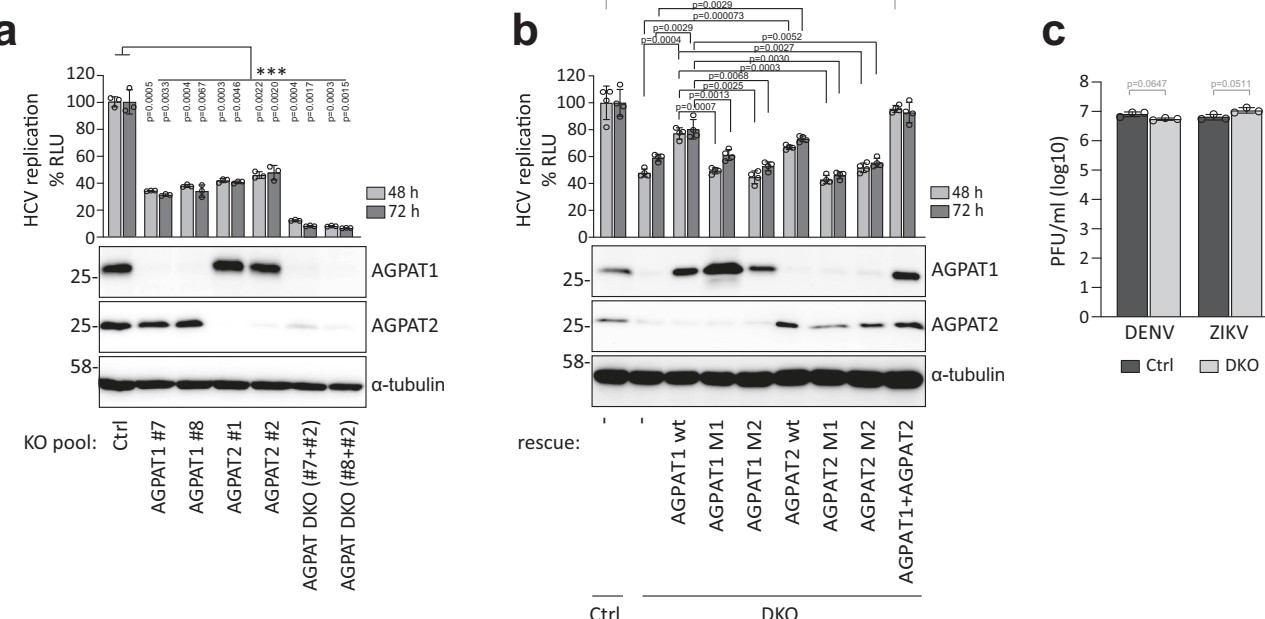

**Fig. 2 Enzymatic activity of AGPATs is required for HCV replication. a** Effect of AGPAT KO on HCV replication. Huh7.5 cells were infected with lentiviruses encoding AGPAT-targeting-sgRNAs and 5 days later, infected with an HCV reporter virus (JcR2a). After 48 h and 72 h, *renilla* luciferase activities in cell lysates, reflecting viral RNA replication, were quantified. Graph shows average and SD from three independent experiments. Significance was calculated by one-tailed paired *t*-test. *p* values are shown in graph. Abundance of AGPAT proteins is shown on the bottom. α-tubulin served as loading control. **b** Enzymatic activity of AGPAT is required for HCV replication. KO cells were reconstituted with sgRNA-resistant AGPAT wild-type (wt) or catalytically inactive mutants (M1 and M2) by lentiviral transduction. Cells were infected with JcR2a, and *renilla* luciferase activities were quantified. Graph shows average and SD from four independent experiments. Significance was calculated by one-tailed paired *t*-test. *p* values are shown in graph. Note the complete rescue by AGPAT1 and 2 co-expression. Abundance of AGPAT proteins is shown below the graph; α-tubulin served as loading control. **c** AGPAT1/2 DKO does not affect DENV or ZIKV propagation. Cells were infected with DENV-2 (strain 16681) or ZIKV (strain H/PF/2013) and 48 h later virus titer was quantified by plaque assay. Graph shows the average and SD from three independent experiments. Significance was analyzed by one-tailed paired *t*-test. *p* values are shown in graph. PFU, plaque-forming units. Source data for panels **a**, **b** and **c** are provided as Source Data file.

interaction with Ras and PA[25]. While in control Huh7 cells this PA sensor displayed a diffuse pattern (Supplementary Fig. 6a), upon co-expression of the HCV NS3-5B polyprotein the sensor accumulated in NS5A-positive puncta (Fig. 4a). Of note, a control PA sensor containing mutations in the PABD of Raf1 (mutant 4E)[26] displayed only a diffuse pattern in NS3-5B expressing cells (Fig. 4a), supporting specificity of the signal and PA recruitment to HCV replication sites. In addition, the wild-type PABD sensor also showed diffuse pattern in NS3-5B expressing AGPAT1/2 DKO cells (Supplementary Fig. 6b).

Since these data suggest an important role of AGPAT1 and 2-dependent PA enrichment on HCV-induced DMVs, we hypothesized that other pathways contributing to PA generation in cells might also play a role in HCV replication. Apart from AGPATs, one other route for PA synthesis is through hydrolysis of phosphatidylcholine (PC) by phospholipase D1 (PLD1) and D2 (PLD2) enzymes (Fig. 4b, top panel)[17,27]. To test the role of PLD1/2 enzymes in HCV replication, we employed a pharmacological approach using 3 different PLD1/2 inhibitors. Treatment with PLD2 inhibitor ML298 caused replication inhibition at a concentration that did not significantly reduce cell viability (~25 μM; Fig. 4b, bottom panel), whereas for the other drugs the reduction in HCV replication correlated with cytotoxicity. To support these findings, we performed siRNA-mediated PLD1/2 depletion in Huh7.5 cells followed by HCV infection. Forty-eight hours post infection, we detected a significant reduction in HCV replication as determined by NS5A-specific immunostaining (Supplementary Fig. 6c). In summary, these results suggest that PA generated via AGPAT1/2, and possibly by alternative PA

synthesis pathway, contributes to HCV replication by supporting the formation of DMVs, which is the site of viral RNA amplification.

**Role of AGPATs and PA in autophagy.** Virus-induced DMVs are morphologically analogous to autophagosomes generated during autophagy[7]; therefore, we tested if PA would be recruited to and is required for autophagy-induced DMVs. To this end, we monitored the localization of the EGFP-tagged PA sensor with markers for DMVs induced during nonselective and selective autophagy. To monitor DMV formation induced during non-selective autophagy, cells were incubated in starvation medium with or without bafilomycin A1 (BafA1), an inhibitor of the vacuolar-type H$^+$-ATPase inducing the accumulation of LC3-positive puncta, which are indicative of autophagosomes. For selective autophagy events, we focused on the induction of DMVs during mitophagy induced by treatment of the cells with valinomycin (Val)[28,29]. As shown in Fig. 4c (top row), the PA sensor EGFP-PABD-Raf1 was rather uniformly distributed throughout the cell in non-induced cells. However, induction of nonselective autophagy by serum starvation led to a significant increase in the number of LC3 puncta with EGFP-PABD-Raf1 relocalizing to these puncta (Fig. 4c). Similarly, induction of mitophagy by Val treatment caused an abundant association of mCherry-Parkin puncta with EGFP-PABD-Raf1 (Fig. 4c, lower panel), whereas in control cells not treated with Val, no such association was found (Supplementary Fig. 6d).

Next, we investigated the role of PA generation during nonselective and selective autophagy. Consistent with the

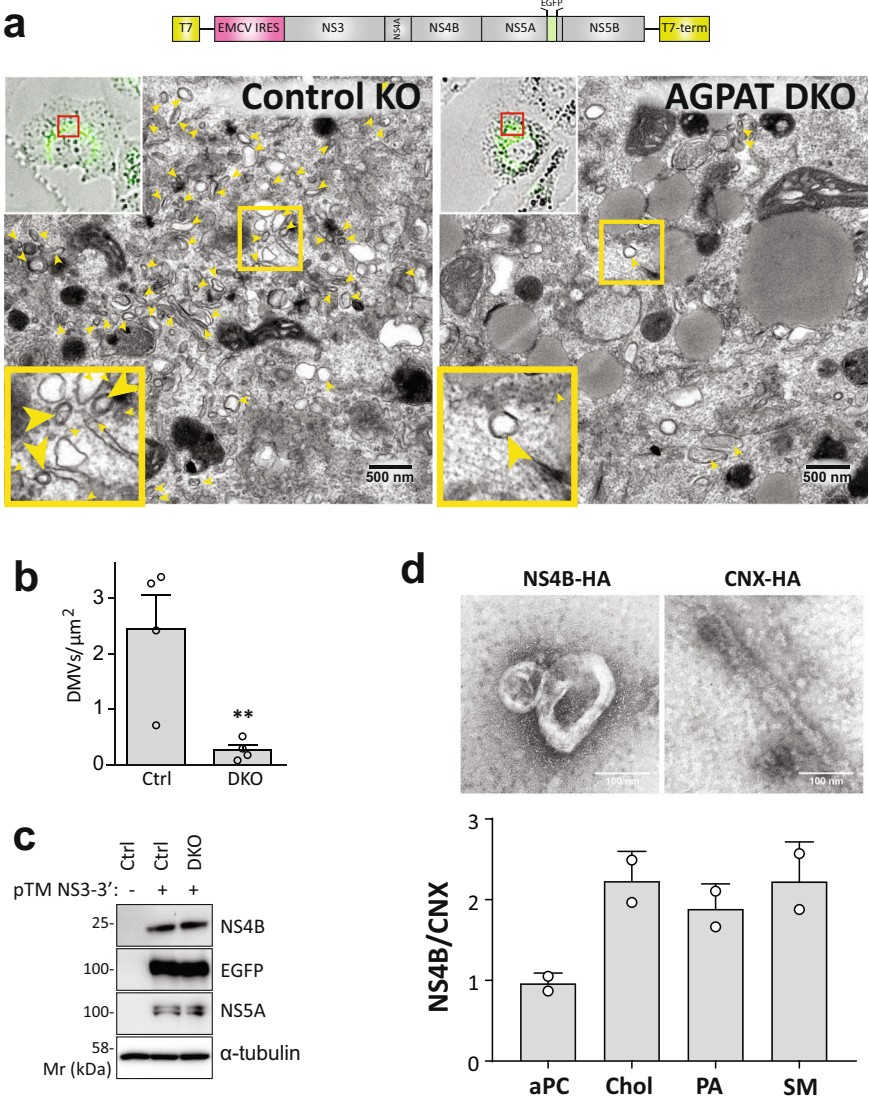

**Fig. 3 Requirement of AGPATs for HCV DMV formation. a–c** AGPAT1/2 DKO dampens DMV formation induced by HCV. Huh7-derived cells stably expressing T7 RNA polymerase and containing or not a double knock-out (DKO) of AGPAT1 and 2 were transfected with HCV replicase-encoding plasmid containing a GFP insertion in NS5A (construct pTM NS3-3'/5A-EGFP, top panel). Transcripts are generated in the cytoplasm via the T7 promoter and terminator (T7-term) sequence and the HCV NS3 – 5B coding region is translated via the IRES of the encephalomyocarditis virus (EMCV). **a** After 24 h, cells were fixed and subjected to CLEM. Low-resolution confocal microscopy images identifying transfected cells are shown on the top left. The area in the red box is shown in the corresponding EM image. Yellow arrow heads indicate DMVs. Insets on the bottom indicate zoomed-in regions. **b** DMVs within whole-cell sections were counted and divided by cell area (µm²). Graph shows average and SD from four different transfected cells. Cells expressing comparable level of HCV replicase were selected for EM analysis. Significance was calculated by two-tailed paired *t*-test. *p* = 0.0128. **c** Expression levels of NS4B and NS5A in transfected cells were determined by western blotting. **d** Lipidome analysis of HCV-induced DMVs. Extracts of Huh7 cells containing the subgenomic replicon sg4BHA31R (NS4B-HA) or stably overexpressing HA-tagged Calnexin (CNX-HA) and control Huh7 cells were used to enrich for DMVs (NS4B-HA) and ER membranes that served as reference (CNX-HA). An aliquot of the sample was analyzed by electron microscopy (top panels) whereas the majority was subjected to lipidome analysis. Representative membrane structures are shown on the top: DMV-like vesicles in the NS4B-HA sample (left) and single membrane tubes in the CNX-HA sample (right). Amounts of selected lipids determined by MS for the NS4B-HA sample were normalized to those obtained for the CNX-HA sample that was set to one (bottom panel). The complete list of analyzed lipids is summarized in Supplementary Data 3. Data are represented as mean ± SD from two biologically independent experiments. Source data for panels **b** and **c** are provided as Source Data file.

relocalization of PA to LC3 puncta during nonselective autophagy, PA inhibitors targeting PLD1, PLD2, and AGPATs, applied as short-term treatments and at non-toxic concentrations, significantly reduced the accumulation of LC3 puncta (Supplementary Fig. 7). These findings are consistent with a recent study suggesting that PA generated on the ATG16L1-positive autophagosome precursor membrane contributes to autophagosome formation[30]. Of note, a third pathway for PA production via

phosphorylation of diacylglycerol (DAG) by diacylglycerol kinase (DAGK)[27], did not contribute to PA accumulation or increase in LC3 puncta during nonselective autophagy (Supplementary Fig. 7).

**Recruitment of AGPATs to SARS-CoV-2 induced DMVs and contribution to viral replication.** Having found that AGPAT1

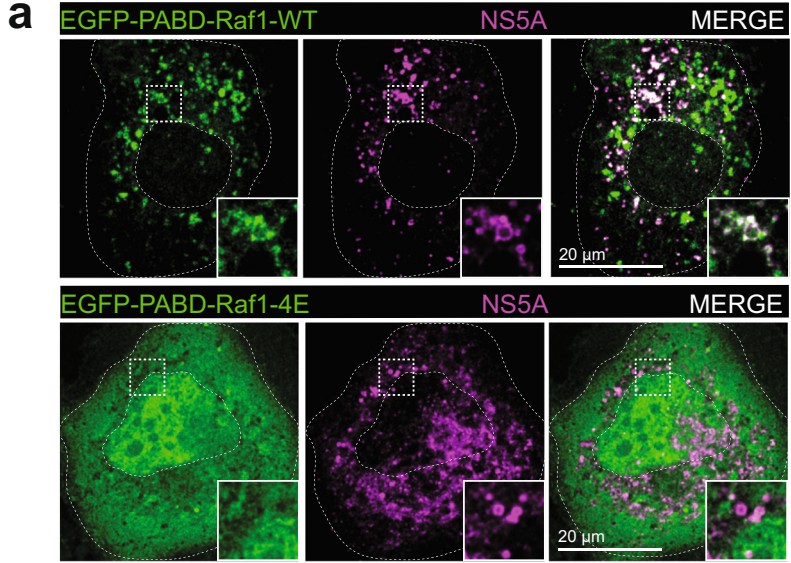

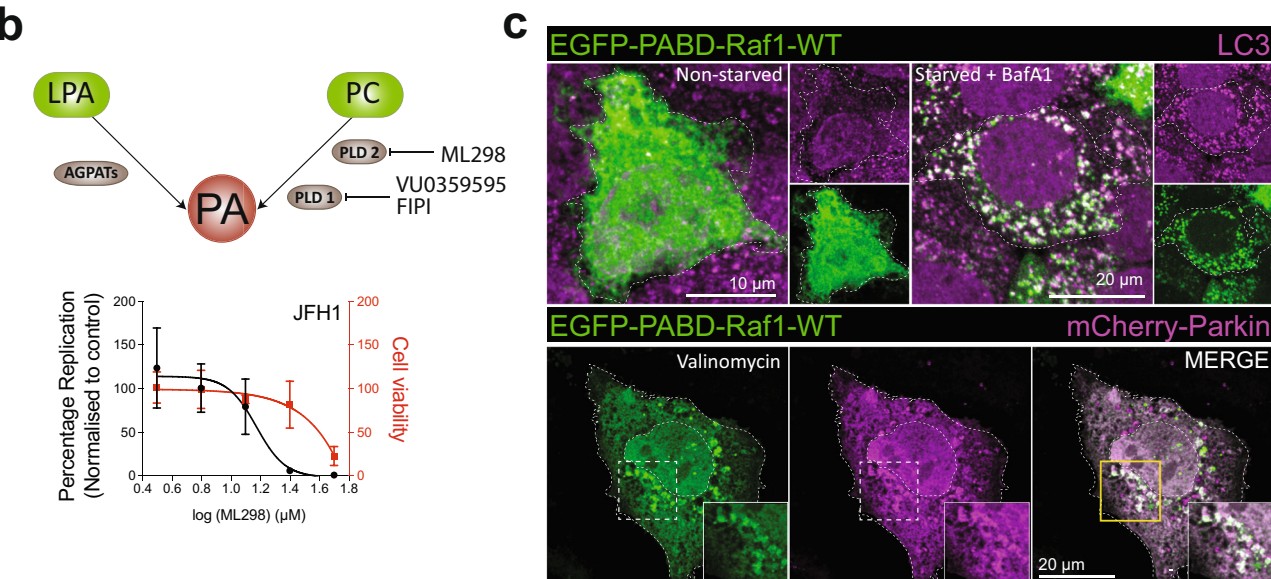

**Fig. 4 PA accumulation on HCV DMVs and autophagy-related structures and role of alternative PA synthesis pathway for HCV replication. a** PA accumulation at NS5A containing structures. Huh7-Lunet/T7 cells were transfected with a construct analogous to the one in Fig. 3a, but containing a mCherry insertion in lieu of EGFP, along with an EGFP-tagged wildtype (WT) or mutant (4E) PA sensor (construct pTM-EGFP-PABD-Raf1-WT or −4E). Twenty-four hours later, EGFP-PABD and NS5A-mCherry were visualized by fluorescence microscopy. White boxes indicate regions magnified in the lower right of each panel. Two biologically independent experiments showed similar results. **b** Top panel: Alternate PA biosynthesis pathways via lysophosphatidic acid (LPA) or phosphatidylcholine (PC) catalyzed by AGPATs or PLDs, respectively. Bottom panel: Huh7-Lunet/T7 cells were electroporated with in vitro transcripts of a subgenomic HCV reporter replicon encoding the firefly luciferase. Four hours after transfection, different concentrations of PA synthesis inhibitors were added to the cells and luciferase activities were analyzed 48 h after electroporation. Graph shows average and SD from three independent experiments. Cell viability determined by CellTiter-Glo luminescent assay is indicated with the red line. Data are represented as mean ± SD from three biologically independent experiments. **c** PA recruitment to autophagy-related structures in selective and non-selective autophagy. Top panel: Huh7-derived cells expressing EGFP-PABD-Raf-1 were incubated in growth medium (top left panels) or in serum-free medium with 200 nM BafA1 (top right panels) for 3 h. Cells were fixed and stained with a LC3 specific antibody. Bottom panel: For selective autophagy, mCherry-tagged Parkin was co-expressed with EGFP-PABD-Raf1, followed by incubation with 10 μM Valinomycin to induce mitophagy. Cells were fixed after 3 h, and EGFP-PABD and mCherry-Parkin were visualized by fluorescence microscopy. Images in panels **e** and **g** are maximum intensity projections. Two biologically independent experiments showed similar results.

and 2, and their reaction product PA, are involved in DMV formation induced upon HCV infection and in, morphologically similar, DMVs generated during autophagy, we hypothesized that AGPATs and PA might also be involved in the biogenesis of replication organelles of other unrelated RNA viruses, e.g., coronaviruses, which also utilize DMVs as viral replication sites[9,10]. Hence, we investigated the role of AGPATs in the DMV biogenesis of SARS-CoV-2, the causative agent of the ongoing

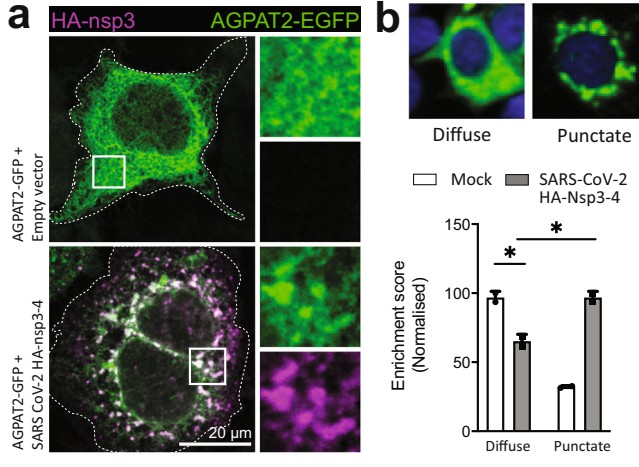

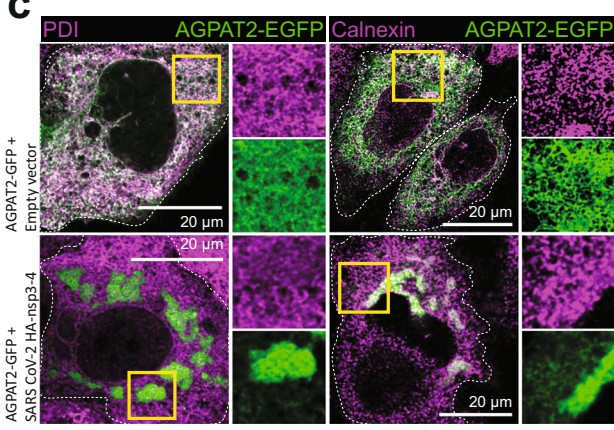

**Fig. 5 Selective recruitment of AGPATs to SARS-CoV-2 induced DMVs. a** Change of subcellular localization of AGPATs upon expression of SARS-CoV-2 nsp3-4. Huh7-derived cells transiently expressing AGPAT2-EGFP were transfected with a SARS-CoV-2 HA-nsp3-4-V5 expression construct or the empty vector. After 48 h, cells were stained with HA-specific antibody and examined by confocal microscopy. Maximum intensity projections are shown. Enrichment score indicates the likelihood of cells showing a punctate or diffuse staining pattern. Two biologically independent experiments showed similar results. **b** Clustering of AGPAT2-EGFP in SARS-CoV-2 HA-nsp3-4-V5 expressing cells. Huh7-Lunet/T7 cells were co-transfected with AGPAT2-EGFP and SARS-CoV-2 HA-nsp3-4-V5 or the empty vector. Twenty-four hours later, cells were fixed and ~1000 cells per condition were separated into two morphotypes (diffuse or punctate) using CellProfiler Analyst-based semi-supervised classifier. Significance was calculated using an ordinary one-way ANOVA. *$p = 0.0211$. A representative experiment from three biologically independent experiments is shown. **c** AGPAT clustering occurs independent of ER remodeling induced by nsp3-4. Huh7-Lunet cells expressing AGPAT2-EGFP and HA-nsp3-4-V5 were stained for the ER markers protein disulfide isomerase (PDI) and calnexin and analyzed by confocal microscopy. Two biologically independent experiments showed similar results. Light microscopy images are maximum intensity projections.

COVID-19 pandemic. In the first set of experiments, we studied the recruitment of AGPATs to SARS-CoV-2 induced DMVs. In the case of MERS-CoV and SARS-CoV, formation of DMVs with structural resemblance to those observed in infected cells can be induced by the sole expression of viral nonstructural protein (nsp)3-4, which is an ~270 kilodalton large polyprotein fragment undergoing self-cleavage[12]. Building on these results we first determined whether the same applies to SARS-CoV-2. Huh7-

derived cells stably expressing T7 RNA polymerase were transiently transfected with a T7 promoter driven SARS-CoV-2 HA-nsp3-4-V5 expression construct or the empty vector (Supplementary Fig. 8a). Using immunofluorescence with an HA-specific antibody, in many cells we observed clusters of HA-nsp3 (Supplementary Fig. 8b). Western blotting confirmed efficient self-cleavage between nsp3 and nsp4 (Supplementary Fig. 8c). To identify membrane alterations in HA-nsp3-4-V5 expressing cells, we employed CLEM. Cells were transfected with the analogous expression construct encoding in addition the mNeonGreen gene to allow visualization of transfected cells by fluorescence microscopy (Supplementary Fig. 8d). mNeonGreen positive cells were recorded and examined by transmission electron microscopy, revealing abundant clusters of DMVs (Supplementary Fig. 8d). Comparison of DMVs induced by nsp3-4 expression and by SARS-CoV-2 infection revealed similar morphology, although expression-induced DMVs appeared to be smaller (~125 nm compared to ~300 nm, respectively) (Supplementary Fig. 8e). These results show that the sole expression of SARS-CoV-2 nsp3-4 is sufficient to induce DMVs with structural similarity to those generated in infected cells.

As the next step, we employed this expression-based system to determine AGPAT function in SARS-CoV-2 nsp3-4 induced DMV formation. Huh7-derived cells expressing EGFP-tagged AGPAT1 or 2 were transiently transfected with the SARS-CoV-2 HA-nsp3-4-V5 encoding plasmid or the empty vector and colocalization of AGPATs with HA-nsp3 was determined by immunofluorescence microscopy. While in empty vector-transfected cells AGPAT2 and 1 were homogeneously distributed throughout the ER (Fig. 5a and Supplementary Fig. 9a, respectively), we observed a strong relocalization of AGPATs in HA-nsp3-4-V5 expressing cells with AGPATs forming puncta that colocalized with HA-nsp3 (Fig. 5a, b and Supplementary Fig. 9a). Of note, the relocalization of AGPATs induced by HA-nsp3-4-V5 was not the result of the massive ER alterations occurring in SARS-CoV-2 infected cells, since the subcellular distribution of other ER-resident proteins, such as protein disulfide-isomerase (PDI) and calnexin remained unaffected compared to the large puncta observed with AGPATs (Fig. 5c).

Since SARS-CoV-2 replication organelles are comprised of DMVs, convoluted membranes and zippered ER[31], we next investigated the membrane structures at the sites of AGPAT colocalization with HA-nsp3-4-V5. Using CLEM, we found that relocalized AGPAT puncta perfectly correlated with extensive networks of SARS-CoV-2 HA-nsp3-4-V5 induced DMVs (Fig. 6a). Overall, these data suggest that similar to HCV, AGPATs are relocalized to SARS-CoV-2 nsp3-4 induced DMVs, the likely sites of viral RNA replication[32].

Next, we tested the effect of AGPAT1/2 depletion on SARS-CoV-2 infection and replication. To this end, we used Huh7-Lunet/T7 DKO cells that were employed for the imaging analyses described so far and stably introduced the SARS-CoV-2 receptor gene *ACE2*. Viral replication was measured by using an image-based assay that quantifies the number of cells containing detectable amounts of the nucleocapsid (N) protein (Supplementary Fig. 9b). Using this approach, we observed significant reduction of SARS-CoV-2 positive cells in both single and double AGPAT KO cells (Fig. 6b, upper panel). Consistently, RT-qPCR revealed similar reduction of viral replication in single and double KO cells (Fig. 6b, lower panel). To exclude a role of AGPAT1/2 in SARS-CoV-2 entry, we used VSVΔG-S pseudotypes and found no effect of AGPAT1/2 KO on entry (Supplementary Fig. 9c, left panel). Consistent with these results, treatment of cells with the general AGPAT inhibitor CI976 (Supplementary Fig. 7a) also did not affect entry of VSVΔG-S pseudotypes (Supplementary Fig. 9c, right panel).

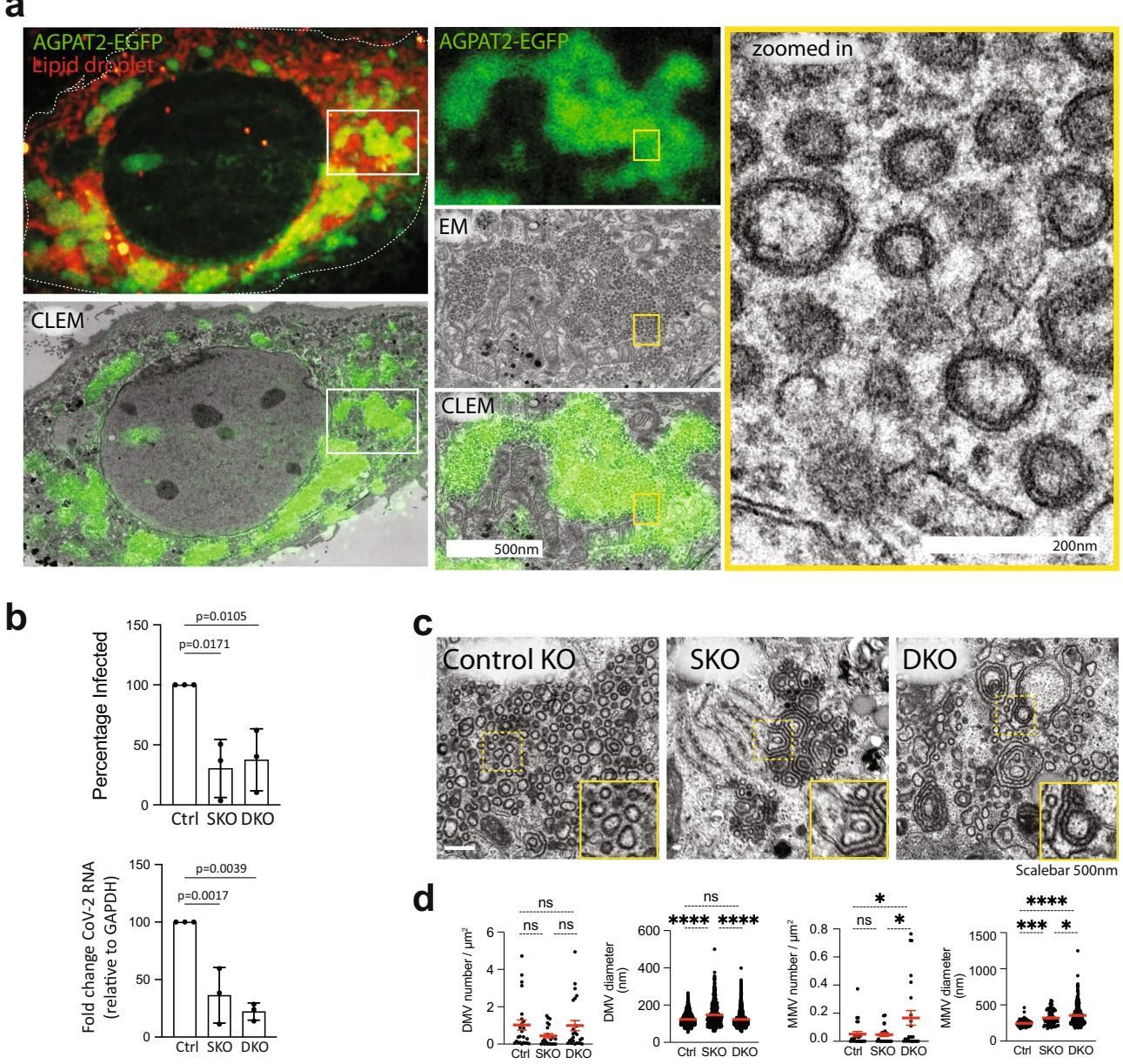

**Fig. 6 Role of AGPATs in SARS-CoV-2 DMV formation and viral replication. a** AGPATs are localized at SARS-CoV-2 HA-nsp3-4-V5 induced DMVs. Huh7-derived cells were transiently transfected with AGPAT2-EGFP HA-nsp3-4-V5 and subjected to CLEM. Light and EM images were correlated by using lipid droplets as fiducial markers. White and yellow boxes indicate areas magnified in the corresponding panels on the right. Several regions from two cells of the same experiment showed similar relocalisation. **b** AGPAT1/2 contribute to SARS-CoV-2 replication. Huh7-Lunet control, AGPAT2 single (SKO) and AGPAT1/2 double (D)KO cells were infected with SARS-CoV-2 (MOI = 0.1). Twenty-four hours later, cells were fixed and immunostained for nucleocapsid, and the percentage of N-positive cells was determined using CellProfiler. Normalized data from three biologically independent experiments are plotted. Significance was calculated using ordinary one-way ANOVA. (top right panel). Total RNA was isolated from infected cells, and SARS-CoV-2 RNA levels were determined using RT-qPCR. Data were normalized to cellular GAPDH mRNA (bottom right panel). Significance was calculated using ordinary one-way ANOVA. Data are represented as mean ± SD from three biologically independent experiments. **c** Aberrant SARS-CoV-2 DMVs in AGPAT1/2 DKO cells. Huh7-Lunet cells with single (SKO) or double knock-out (DKO) and stably expressing T7 polymerase were transfected with a plasmid encoding SARS-CoV-2 HA-nsp3-4-V5 and fluorescent neon-green. Twenty-four hours later, cells were fixed and NeonGreen positive cells were recorded and examined by EM. Scalebar 500 nm. **d** HA-nsp3-4-V5 induced DMVs and multi-membrane vesicles (MMVs) were quantified. Shown are the number and diameter of DMVs and MMVs in these cells as observed from n = 8 cell profiles per condition. Statistical significance was calculated using ordinary one-way ANOVA. ****p = 1.67453E-49. Data are represented as mean ± SD. Light microscopy image in panel **a** is a maximum intensity projection.

To determine if reduced SARS-CoV-2 replication in AGPAT1/2 KO cells might correlate with altered DMV formation, we transiently expressed SARS-CoV-2 HA-nsp3-4-V5 in control, single and double KO cells. The absence of AGPAT 1/2 did not significantly affect the abundance of cleaved viral proteins HA-

nsp3 and nsp4-V5 (Supplementary Fig. 8c). EM analysis of control cells revealed HA-nsp3-4-V5 induced membrane alterations, consistent with an earlier report for MERS-CoV and SARS-CoV[12] (Fig. 6c, d). This included zippered ER and DMVs with an average diameter of ~145 nm. In contrast to HCV, the number of

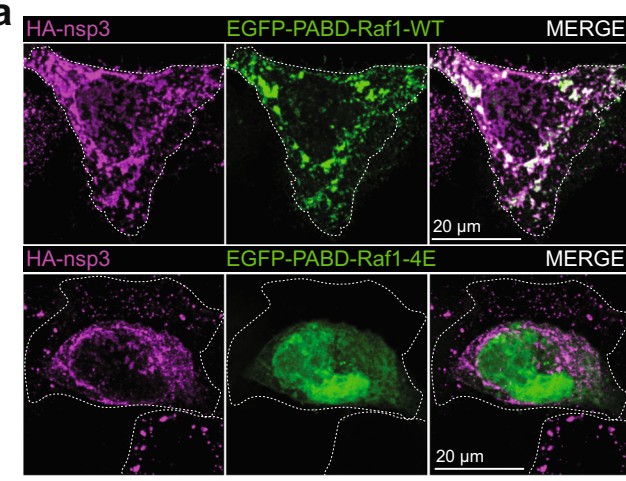

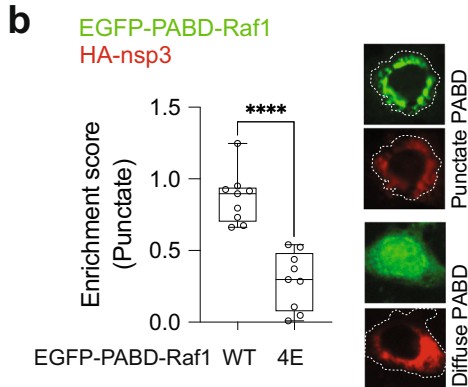

**Fig. 7 PA accumulation at SARS-CoV-2 DMVs. a** PA enrichment at SARS-CoV-2 nsp3-containing structures. Huh7-Lunet cells expressing the wildtype or mutant form of the PA sensor were transfected with the plasmid encoding HA-nsp3-4-V5 and 24 h later, cells were fixed, immunostained with HA-specific antibody and HA-nsp3 and EGFP-PABD were visualized by confocal microscopy. Maximum intensity projections are shown. **b** Using CellProfiler Analyst, a semi-supervised machine-learning classifier was trained to differentiate between punctate and diffuse signals of the EGFP-PABD sensor (top panel). A normalized enrichment score which indicates the probability of cells showing punctate EGFP-PABD localization to nsp3 fluorescent signal across the whole-cell population is shown in the graph on the bottom panel as a scatterplot. The central line of the plot indicates median with first and third quartiles are shown as boxes, and whiskers as lines. Significance was calculated by unpaired two-tailed *t*-test. ****$p = 7.39734E\text{-}6$. Data are represented as mean ± SD from three biologically independent experiments showing three technical replicates from each experiment.

nsp3-4 induced DMVs did not decrease in AGPAT single and double KO cells (Fig. 6d, left two panels). However, in both cell pools we observed marked accumulations of multi-membrane vesicles (MMVs), indicating the formation of aberrant membrane structures (Fig. 6c, d).

**Accumulation of PA at SARS-CoV-2 DMVs and role of alternative PA synthesis pathways**. To test whether similar to AGPAT1/2 relocalization to nsp3-4 induced DMVs, PA is also enriched at those sites we used the EGFP-tagged PA sensor derived from Raf1. In Huh7-derived cells expressing SARS-CoV-2 HA-nsp3-4-V5, the functional version of the sensor (EGFP-PABD-Raf1-WT) strongly colocalized with HA-nsp3 in distinct

puncta, whereas no such puncta were found with the mutant PABD-Raf1, confirming specificity of PA sensor recruitment to HA-nsp3-containing sites (Fig. 7a, b).

Although in comparison to HCV, AGPAT1/2 DKO had lower impact on SARS-CoV-2 replication (compare Fig. 2a with Fig. 6b), and caused a morphologically distinct phenotype of nsp3-4 induced DMVs (Figs. 3a and 6c, respectively), AGPATs, and most likely PA, still accumulated at sites of SARS-CoV-2 DMV clusters (Fig. 7a, b). This indicates that PA synthesis pathways other than via AGPAT1/2, might contribute to SARS-CoV-2 replication and DMV formation. By means of pharmacological inhibition of enzymes that convert LPA, PC, and DAG to PA (Supplementary Fig. 7a), we measured the dose-dependent effect of these drugs on SARS-CoV-2 replication. All inhibitors reduced SARS-CoV-2 replication in Calu-3 cells, and in A549 cells stably expressing ACE2, two commonly used cell models for this virus, at non-cytotoxic concentrations, although in the case of the general AGPAT inhibitor CI976 selectivity was rather low (Fig. 8a and Supplementary Fig. 10a, respectively). Of note, combining the inhibitors at concentrations close to or below their IC50 values caused much stronger reduction of virus replication with no or minimal effect on cell viability, indicating that SARS-CoV-2 can utilize PA produced by alternative PA synthesis pathways (Supplementary Fig. 10a, b). We then measured the effect of these drugs on PA accumulation at HA-nsp3 containing puncta in HA-nsp3-4-V5 expressing cells and found that all inhibitors reduced PA levels at these sites (Fig. 8b). This reduction was not the result of altered HA-nsp3-4-V5 expression level or self-cleavage, which were unaffected in inhibitor-treated cells (Supplementary Fig. 10c). Next, we determined if reduced PA levels caused by these inhibitors also affect SARS-CoV-2 nsp3-4 induced DMV formation. In cells treated with AGPAT, PLD1, and DAGK inhibitors DMV diameters were significantly reduced (Fig. 8c, d). Moreover, PLD2 inhibition promoted the formation of MMVs and larger DMVs, similar to what we found in AGPAT single and double KO cells (Fig. 6c). Taken together, our data suggest that PA enrichment is important for proper SARS-CoV-2 DMV formation and viral replication.

## Discussion

Here, we show that PA produced by AGPAT1 and 2 is important for the replication of evolutionary distant positive-strand RNA viruses, HCV and SARS-CoV-2 that amplify their genome in association with DMVs. The remarkable dependence on a common host lipid for proper DMV biogenesis induced by these phylogenetically disparate viruses indicates a striking commonality in the biogenesis of their replication organelles. Conversely, for viruses replicating their RNA genome in ER-derived membrane invaginations such as the flaviviruses DENV and ZIKV, this lipid pathway appears to be dispensable[4,33]. Of note, PA production through AGPAT1 and 2 is also involved in the formation of autophagosome-like DMVs, arguing for some similarity between cellular and viral DMV formation and lipid composition. In addition, alternative routes of PA biosynthesis contribute to HCV and SARS-CoV-2 replication and DMV generation. These alternative routes might complement each other, which would explain the differential effects we observed for PLD1 and PLD2 inhibitors (Fig. 8a, b).

It has been shown that increase in the cellular concentration of PA and signaling via PA activates mTOR, which in turn inhibits autophagy[34]. At first sight, this is counterintuitive, because induction of autophagy by HCV has been reported[35], while in the present study we show that HCV requires PA for efficient replication. However, we note that although HCV activates autophagy, it is not required per se for viral replication, but rather

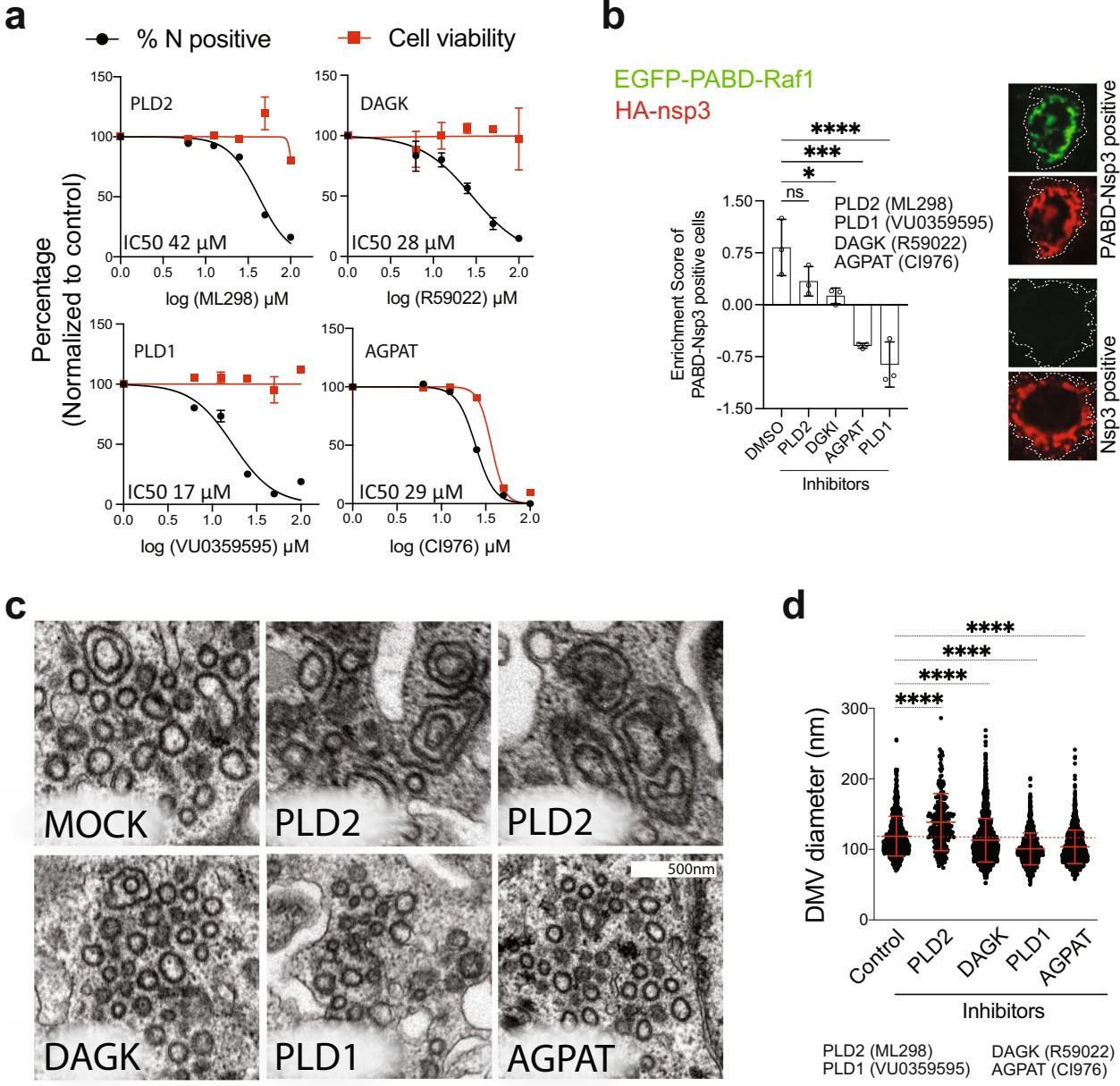

**Fig. 8 Role of alternative PA synthesis pathways for SARS-CoV-2 replication and DMV formation. a** Alternative PA synthesis pathways are important for SARS-CoV-2 replication. Calu-3 cells were infected with SARS-CoV-2 (MOI = 5) in the presence of AGPAT, PLD1/2, or DAGK inhibitors. Cells were fixed 24 h post infection, stained with nucleocapsid-specific antibody, and the percentage of infected cells was quantified using CellProfiler. Cell viability and percentage inhibition are plotted as dose-response curves and IC50 values are given on the top of each panel. Data are represented as mean ± SD from a representative experiment; two biologically independent experiments gave similar results. **b** AGPAT, PLD and DAGK inhibitors reduce PA accumulation at nsp3-positive structures. Huh7-Lunet cells were transfected with SARS-CoV-2 HA-nsp3-4-V5 and EGFP-PABD-Raf1 encoding plasmids, followed by the addition of a given inhibitor 4 h after transfection. Twenty-four hours later, cells were fixed and HA-nsp3 was detected with an HA-specific antibody. EGFP-PABD and HA-nsp3 were visualized by confocal microscopy. A semi-automated machine-learning-based classifier was trained to separate HA-nsp3/PABD double-positive structures from HA-nsp3 single positive structures. Enrichment score for HA-nsp3/PABD double-positive structures showing the up or downregulation of double-positive cells in different samples is plotted and statistical significance was calculated using ordinary one-way ANOVA. ns, $p = 0.1231$, *$p = 0.0241$, ***$p = 0.0002$, ****$p = 6.45177E-5$. Data are represented as mean ± SD from three biologically independent experiments. **c** Decrease of SARS-CoV-2 DMV diameter by AGPAT, PLD, and DAGK inhibitors. Huh7-Lunet/T7 cells were transfected with the plasmid encoding HA-nsp3-4-V5 and fluorescent NeonGreen, followed by addition of inhibitors 4 h after transfection. Twenty-four hours later, cells were fixed, NeonGreen positive cells were recorded and examined by EM. Representative images are shown for each condition. **d** Number and morphology of DMVs were determined for at least seven cell profiles per condition. DMV diameters are plotted and statistical significance was calculated using ordinary one-way ANOVA. ****$p = 4.03773E−25$. Data are represented as mean ± SD.

distinct components of the autophagy machinery[36]. Moreover, the effect of HCV infection on mTOR activation is discussed controversially[37,38]. Importantly, our data argue for local production of PA at viral replication sites via the recruitment of AGPAT proteins, and therefore, mTOR might be activated only locally at these sites, but not at the whole-cell level.

At least three possibilities can be envisioned how PA promotes DMV formation in viral replication and in the context of autophagy. First, the presence of lipids with cone or inverted cone shape in membranes contributes to membrane bending by generating negative or positive membrane curvature, respectively[16]. While LPA has a large polar head group to fatty acid tail ratio, giving rise to an inverse-cone shape and resulting in positive membrane curvature, the additional fatty acid tail present in PA inverses the head-to-tail ratio. Hence PA displays an overall cone shape, which contributes to negative membrane curvature. Thus, the LPA–PA conversion by AGPATs might contribute to DMV formation by facilitating membrane curvature. Second, PA is directly or indirectly implicated in membrane fission[39]. This might be achieved by recruitment of effector proteins by PA, either through downstream signaling events, or directly by serving as docking site for PA-binding proteins that have amphipathic or hydrophobic surfaces. In this regard, our NS4B-proteome showed the enrichment of three known PA-interacting proteins, namely, Vitronectin, splicing factor-1, and ubiquitin carboxy-terminal hydrolase L1, in the viral DMV fraction (Supplementary Data 1)[40]. More than 50 different proteins have been reported to interact with PA, including protein kinases, phosphatases, nucleotide-binding proteins and regulators, however, a comprehensive list remains elusive, and their possible role in the formation of DMVs during autophagy or viral RNA replication, if any, remains to be determined[18]. Third, an additional role of PA for the functionality of viral replication organelles and perhaps also autophagosomes might be in serving as an exchange lipid in a counter-transporter chain as suggested earlier[7,41]. In the case of HCV, we and others identified accumulation of PI4P at DMVs (reviewed in[4,5]) and similar findings have been made for membranous structures involved in the early steps of autophagy[42]. For HCV, it is thought that PI4P recruits lipid transporters such as OSBP that deliver cholesterol into DMV membranes in exchange for PI4P. A similar mechanism might apply for other lipids or the PI4P precursor PI, with PA serving as a possible exchange factor against these other lipids or PI, respectively.

The similar dependency of DMV-type replication organelles on PA, as reported here for HCV and SARS-CoV-2, might offer an attractive starting point for broad-spectrum antivirals targeting a diverse range of positive-strand RNA viruses replicating in such structures. In line with this assumption, an inhibitor of cytosolic phospholipase $A_2\alpha$ has been reported to suppress replication and DMV formation of the 229E human coronavirus and to exert antiviral activity also against the alphavirus Semliki forest virus[43]. In addition, several human diseases have been linked to defects in PA metabolism and selective autophagy, including neurological disorders and chronic obstructive pulmonary disease[18,19]. Although the precise role of PA in these diseases remains to be determined, the critical role of PA for HCV and SARS-CoV-2 infection reported here might offer new approaches for therapeutic intervention.

## Methods

**Plasmids**. To construct the lentiviral vectors pWPI-EGFP-CT, pWPI-EGFP-NT and pWPI-mCherry-NT, EGFP or mCherry coding sequences were amplified by PCR and inserted by in-fusion reaction into the linearized pWPI vector using the BamHI restriction site. Lentiviral plasmids pWPI-AGPAT1-EGFP and pWPI-AGPAT2-EGFP were constructed by insertion of the human AGPAT1 gene (gene ID: 10554) or human AGPAT2 gene (gene ID: 10555) into the BamHI site of the pWPI-EGFP-CT vector. To construct the expression vector pGEX-PABD and

pGEX-PABD_4E, the PA-binding domain (PABD) of the yeast Spo20 gene (gene ID: 855031) was used. The nucleotide sequence of the PABD is as follows: gacaattgttcaggaagcagaagacgtgataggctacatgtgaagcttaaatccttgaggaataaaatccacaaa-caacttcacccaaactgtcggttcgatgacgccactaagactagt. The 4E mutant PABD contained K66E, K68E, R71E, and K73E substitutions as described previously[24]. PABD_WT or PABD 4E mutant sequences were amplified by PCR and inserted into the pGEX-6P-1 vector using BamHI and XhoI restriction sites. The PABD of Raf1 (gene ID: 5894), which corresponds to amino acid residue 390-426 of the Raf1 protein, was amplified by PCR using the Addgene plasmid 116785 as the source for the DNA. The amplicon was inserted into the pWPI-EGFP-NT vector using HiFi assembly of DNA fragments. The 4E Raf1 PABD mutant contains the following amino acid substitutions as reported earlier: R391E, R398E, K399E, and R401E[26]. The human Parkin gene (gene ID: 5071) was amplified by PCR and inserted into the pWPI-mCherry-NT vector to obtain pWPI-mCherry-Parkin. The PABD T7 expression constructs were generated using the same approach and the sequences were inserted into pTM1-2eGFP vector to obtain pTM-PABD-Raf1-WT and pTM-PABD-Raf1-4E constructs. All plasmids used in this study are listed in Supplementary Table 1.

**Reagents and resources**. All reagents and resources as well as antibodies used in this study are listed in Supplementary Tables 2 and 3, respectively. Phorbol 12-myristate 13-acetate (PMA), Bafilomycin A1 (BafA1), Valinomycin (Val) and Leu-Leu methyl ester hydrobromide (LLOMe) were dissolved in DMSO to prepare stock solutions.

**Cell culture and transfection**. All cell lines used in this study are listed in Supplementary Table 4. Cells were maintained in Dulbecco's modified Eagle medium (DMEM) (Thermo Fisher Scientific), supplemented with 2 mM L-glutamine, nonessential amino acids, 100 U/ml penicillin, 100 μg/ml streptomycin, and 10% fetal calf serum (DMEM cplt). To select for transduced cells, they were cultured in medium containing antibiotics as specified in Supplementary Table 4. To trigger and visualize starvation-induced autophagy, cells were incubated with serum and amino acid deprived DMEM with or without 200 nM BafA1 for 3 h at 37 °C. To monitor mitophagy events, cells were treated with medium containing 10 μM valinomycin at 37 °C. To induce PA redistribution, cells were incubated with 100 nM PMA for 5 min. For DNA transfection we used TransIT-LT1 Transfection Reagent according to the manufacturer's protocol (Mirus Bio LLC). For RNA electroporation, $4 \times 10^6$ cells were suspended in 400 μl cytomix containing 5 μg in vitro transcript, 5 mM glutathione and 2 mM ATP. Electroporation was performed using a Gene Pulser system (Bio-Rad) at 975 μF and 270 V in a 0.4-cm cuvette (Bio-Rad).

**Lentivirus production and transduction of cells**. Lentivirus production and cell transductions were performed exactly as described earlier[44]. In brief, HEK-293T cells were co-transfected with the packaging plasmid pCMV-dR8.91, the envelope-encoding plasmid pMD2.G and a pWPI vector plasmid containing the gene-of-interest by use of polyethylenimine (Polysciences Inc.). Supernatants were harvested 48 and 72 h post-transfection, filtered and virus titers were determined by colony formation assay.

**HCV production and viral infection**. Production of HCV stocks and infection of cells were performed as described recently[45]. In brief, Huh7.5 cells were transfected with in vitro transcripts of the HCV variant Jc1 or a renilla-luciferase encoding variant thereof (JcR2A) by electroporation. After 24 h, supernatants were replaced with fresh medium and 48 and 72 h post-electroporation, supernatants were collected and filtered through 0.45-μm-pore-size filters. Supernatants were stored at −70 °C prior to the determination of virus titers by limiting dilution assay on Huh7.5 cells. Infected cells were detected by immunohistochemistry using the NS5A-specific antibody 9E10, and the 50% tissue culture infective dose (TCID50) was determined.

**Dengue virus and zika virus production**. The reporter virus genomes of DENV-2 (strain 16681 s; DV-R2A) and ZIKV (strain H/PF/2013; synZIKV-R2A), which encode renilla luciferase, have been previously described[46,47]. DENV stocks were prepared by electroporation of BHK-21 cells with DV-R2A in vitro transcripts and amplified in VeroE6 cells. SynZIKV-R2A stocks were prepared by electroporation of VeroE6 cells or insect C6/36 cells. Extracellular virus titers were determined by plaque-forming unit (PFU) assay in VeroE6 cells using an overlay medium containing 1.5% carboxymethylcellulose.

**SARS-CoV-2 production**. The SARS-CoV-2 isolate Bavpat1/2020 was kindly provided by Christian Drosten (Charité Berlin, Germany) through the European Virology Archive (Ref-SKU: 026V-03883) at passage 2. Virus stocks were produced in VeroE6 cells by passaging the virus two times. Titers of infectious virus were determined by plaque assay as reported earlier[48].

**Generation of knockout cell lines**. The 20-base pairs long guide strands used to target AGPATs are listed in Supplementary Table 5. CRISPR plasmids were

constructed by insertion of annealed oligonucleotides into the lentiCRISPRv2 plasmid (Addgene) encoding a puromycin resistance gene in the case of AGPAT1 or into the lentiCRISPR plasmid (Addgene) encoding a blasticidin resistance gene in the case of AGPAT2. To generate knockout cell lines, cells were transduced with a given lentivirus and two days later cells were cultured in medium containing 3 μg/ml puromycin or blasticidin for at least 3 days. Knock-out was validated by Western blot.

**Purification of NS4B-associated membranes for proteome analysis.** The protocol used to purify membrane fractions was derived from the one we have reported earlier[13] and modified in order to increase yields. This became possible by using the MACS purification system (Miltenyi Biotec) that was however, incompatible with fractions prepared by sucrose gradient centrifugation because of high viscosity. Huh7-Lunet cells (~$2.5 \times 10^8$) containing the subgenomic replicon sg4B[HA]31R, and cells containing the analogous replicon with non-tagged NS4B and control Huh7-Lunet cells stably overexpressing HA-tagged Calnexin (CNX-HA) were used. Cells were washed twice with PBS, scraped into PBS and pelleted by centrifugation at $1400 \times g$ for 2 min at room temperature. Cells were resuspended in 4 ml hypotonic buffer (20 mM Tris [pH 8.0], 1.5 mM MgCl$_2$, 10 mM Na-acetate) and incubated on ice for 30 min. Cells were divided into two tubes and disrupted by pressuring 25 times through 2-ml syringes fitted with 22-gauge needles. Nuclei and cell debris were removed by centrifugation at $800 \times g$ for 10 min at 4 °C. Post-nuclear supernatants were equilibrated to 150 mM NaCl and subjected to HA-affinity purification. One hundred ul slurry of HA-antibody coated magnetic μMACS beads (Miltenyi Biotec) were incubated with post-nuclear supernatants on a rotator for 2 h at 4 °C. Beads were bound to MidiMACS columns and washed 5 times with 5 ml IP buffer (20 mM Tris [pH 8.0], 150 mM NaCl, 1.5 mM MgCl$_2$, 10 mM Na-acetate). Bound proteins were eluted from the beads by using SDS buffer (50 mM HEPES [pH 7.9], 150 mM NaCl, 5 mM EDTA, 2% SDS). Eluates were subjected to filter aided sample preparation by using a 3 kDa molecular weight cutoff filter (VIVACON 500; Sartorius Stedim Biotech GmbH, 37070 Goettingen, Germany) according to the procedure described earlier[49]. Fifty microliters of sample were directly mixed in the filter unit with 200 μl of freshly prepared 8 M urea in 100 mM Tris-HCl [pH 8.5] (UA buffer) and centrifuged at 14,000 × g for 15 min at 20 °C to remove SDS. Any residual SDS was washed out by two washing steps with 200 μl UA buffer. Proteins were alkylated by incubation with 100 μl 50 mM iodoacetamide in the dark for 30 min at room temperature. After washing three times with 100 μl of UA buffer and three times with 100 μl of 50 mM triethylammonium bicarbonate (TEAB) buffer [pH 8.0] (SIGMA-Aldrich Chemie GmbH, Germany), proteins were digested with 1.25 μg trypsin overnight at 37 °C. Peptides were recovered from the filter by centrifugation, applying 40 μl of 50 mM TEAB buffer, followed by 50 μl 0.5 M NaCl (SIGMA-Aldrich Chemie GmbH). Eluted peptides were acidified with TFA, desalted using C18 solid-phase extraction spin columns (The Nest Group, Southborough, MA), organic solvent removed in a vacuum concentrator at 45 °C and reconstituted in 5% formic acid for analysis by liquid chromatography coupled to tandem mass spectrometry (LC-MS/MS).

LC-MS/MS was performed on a hybrid linear trap quadrupole (LTQ) Orbitrap Velos mass spectrometer (ThermoFisher Scientific, Waltham, MA, USA) coupled to an Agilent 1200 HPLC nanoflow system (Agilent Biotechnologies, Palo Alto, CA, USA) via nanoelectrospray ion source using a liquid junction (Proxeon, Odense, Denmark). Normalized amounts of tryptic peptides (~1 μg) were loaded onto a trap column (Zorbax 300SB-C18 5 μm; 5 × 0.3 mm; Agilent Biotechnologies) at a flow rate of 45 μl/min using 0.1% TFA as loading buffer. After loading, the trap column was switched in-line with a 75 μm inner diameter, 25 cm long analytical column (packed in-house with ReproSil-Pur 120 C18-AQ, 3 μm; Dr. Maisch, Ammerbuch-Entringen, Germany). Mobile-phase A consisted of 0.4% formic acid in water and mobile-phase B of 0.4% formic acid in a mix of 90% acetonitrile and 9.61% water. The flow rate was set to 230 nl/min and a 90 min gradient applied (36–30% solvent B within 81 min, 30–65% solvent B within 8 min, 65–100% solvent B within 1 min, 100% solvent B for 6 min before equilibrating at 36% solvent B for 18 min). For the MS/MS experiment, the LTQ Orbitrap Velos mass spectrometer was operated in data-dependent acquisition (DDA) mode with the 15 most intense precursor ions selected for collision-induced dissociation (CID) in the linear ion trap (LTQ). MS1-scans were acquired in the Orbitrap mass analyzer using a scan range of 350–1800 m/z at a resolution of 60,000 (at 400 m/z). Automatic gain control (AGC) was set to a target of $1 \times 10^6$ and a maximum injection time of 500 ms. MS2-scans were acquired in parallel in the linear ion trap with AGC target settings of $5 \times 10^4$ and a maximum injection time of 50 ms. Precursor isolation width was set to 2 Da and the CID normalized collision energy to 30%. The threshold for selecting precursor ions for MS2 was set to ~2000 counts. Dynamic exclusion for selected ions was 30 s. A single lock mass at m/z 445.120024 was employed[50].

**Proteome MS data analysis.** Acquired raw data files were processed using the Proteome Discoverer 2.2.0.388 platform, utilizing the database search engine Sequest HT. Percolator V3.0 was used to remove false positives with a false discovery rate (FDR) of 1% on peptide and protein level under strict conditions. Precursor masses were recalibrated prior to Sequest HT searches using full tryptic digestion against the human SwissProt database v2017.06 (20,456 sequences and appended known contaminants) with up to one miscleavage site. Oxidation (+15.9949 Da) of methionine and acetylation (+42.010565 Da) of protein N-terminus were set as variable modifications, whilst carbamidomethylation (+57.0214 Da) of cysteine residues was set as fixed modifications. Data was searched with mass tolerances of ±10 ppm and 0.6 Da on the precursor and fragment ions, respectively. Results were filtered to include peptide spectrum matches (PSMs) with Sequest HT cross-correlation factor (Xcorr) scores of ≥1. For calculation of protein intensities, the Minora Feature Detector node and Precursor Ions Quantifier node, both integrated in Thermo Proteome Discoverer were used. Automated chromatographic alignment and feature linking mapping ("matching between run"; https://www.maxquant.org) were enabled. Precursor abundance was calculated using intensity of peptide features including only unique peptide groups.

SAINTexpress version 3.6.3[51] was used to identify interactors in NS4B and CNX pulldowns. The protein intensities obtained from Proteome Discoverer analysis were averaged over technical replicates and used as inputs for the analysis with the SAINTexpress tool. Proteins having SAINT AvgP > 0.95 were considered as interactors. All proteins identified by SAINT as interactors (1543 proteins) of at least one of the baits (NS4B or CNX) were normalized to equal total abundance in each sample analyzed for differential abundance between the NS4B and CNX pulldowns using the limma R software package[52]. Functional networks for 309 NS4B and 195 CNX significantly enriched proteins (twofold enriched and having Limma q values of ≤0.05) were generated using ClueGO v2.5.5 app embedded in Cytoscape 3.7.2[53]. The human GO (Biological Processes, version from 27 February 2019) was used with the following settings: type of analysis: single; GO terms level: 3–4; GO term restriction: 3 genes and 4%; evidence code: all experimental. A significance threshold level of 0.05 was applied (Supplementary Data 1).

**Purification of NS4B-associated membranes and lipid analysis.** For lipid analysis, NS4B-associated membranes were purified as described above with the following modifications. HA-antibody coated magnetic beads (Thermo Scientific) were used as recommended by the manufacturer and bound material was eluted with 50 μl 0.1 M glycine [pH 2.5] for 10 min at room temperature followed by neutralization with 30 μl of 1 M Tris [pH 7.5]. Eluates of split samples were pooled and subjected to negative staining for quality control prior to lipidomic analysis.

For lipidome analysis, membranes released from HA-beads were subjected to an acidic Bligh and Dyer extraction using chloroform/methanol/37% HCl (40:80:1, vol:vol:vol) as previously described[54]. Extractions were performed in the presence of a lipid standard mix containing 25 pmol phosphatidylcholine (13:0/13:0, 14:0/14:0, 20:0/20:0; 21:0/21:0; Avanti Polar Lipids, Alabaster, AL, USA), 25 pmol sphingomyelin (d18:1 with N-acylated 13:0, 17:0, 25:0, semi-synthesized as described in[54], 50 pmol D6-cholesterol (Cambridge Isotope Laboratory), 15 pmol phosphatidylinositol (16:0/ 16:0 and 17:0/20:4; Avanti Polar Lipids), 12.5 pmol phosphatidylethanolamine, 12.5 pmol phosphatidylserine and 5 pmol phosphatidylglycerol (all 14:1/14:1, 20:1/20:1, 22:1/22:1, semi-synthesized as described in[54], 12.5 pmol diacylglycerol (17:0/17:0, Larodan), 12.5 pmol cholesteryl ester (9:0, 19:0, 24:1, Sigma-Aldrich, St. Louis, MO, USA), 12 pmol triacylglycerol (LM-6000/D5-17:0/17:1/17:1; Avanti Polar Lipids), 2.5 pmol ceramide and glucosylceramide (both d18:1 with N-acylated 15:0, 17:0, 25:0, semi-synthesized as described in[54], 2.5 pmol lactosylceramide (d18:1 with N-acylated C12 fatty acid; Avanti Polar Lipids), 2.5 pmol phosphatidic acid (17:0/20:4, Avanti Polar Lipids) and 2.5 pmol lyso-phosphatidylcholine (17:1; Avanti Polar Lipids). Lipid extracts were resuspended in 60 μl methanol and samples were analyzed on an AB SCIEX QTRAP 6500+ mass spectrometer (Sciex, Framingham, MA, USA) with chip-based (HD-D ESI Chip; Advion Biosciences, Ithaca, NY, USA) nano-electrospray infusion and ionization via a Triversa Nanomate (Advion Biosciences) as previously described[54]. Resuspended lipid extracts were diluted 1:10 in 96-well plates (Eppendorf twin tec 96, colorless, Z651400-25A; Sigma-Aldrich, St. Louis, MO, USA) prior to measurement. Lipid classes were analyzed in positive ion mode applying either specific precursor ion (PC, lyso- PC, SM, cholesterol, Cer, HexCer, and Hex2Cer) or neutral loss (PE, PS, PI, PG, and PA) scanning as described in ref.[54].

Data evaluation was performed using LipidView (RRID: SCR_017003; Sciex, Framingham, MA, USA) and an in-house-developed software package (ShinyLipids).

**siRNA screening.** siRNA screening was performed by solid-phase reverse transfection of Huh7.5 cells seeded into 96-well plates as described previously[55]. In brief, siRNA and transfection reagent were seeded into each well of a 96-well plate. After air drying of the plates, $5 \times 10^3$ Huh7.5 cells stably expressing firefly luciferase (Fluc) were seeded per well in a volume of 200 μl. After 3 days, cells were infected with the renilla luciferase (Rluc) HCV reporter virus JcR2A[55]. To determine the impact of knockdown on HCV entry and replication, cells were lysed 72 h after infection and renilla luciferase activity was measured. To account for potential cytotoxic effects of siRNAs, FLuc was measured in the same lysate by using dual luciferase assay. Statistical analysis of the siRNA screening data was performed in R version 3.4, using the Bioconductor package RNAither[56]. In brief, data were quality-checked, Rluc values were then log-transformed and Lowess normalized against FLuc measurements to account for cytotoxic effects. Measurements were then normalized using z-score normalization with regard to the negative controls, and replicates summarized using the mean.

**Luciferase reporter assay**. Cells were lysed in 200 μl luciferase lysis buffer (1% Triton X-100, 25 mM glycyl glycin [pH 7.8], 15 mM $MgSO_4$, 4 mM EGTA, 10% glycerol) per well in a 12-well plate. Plates were stored at −20 °C until measurement of luciferase activity. For firefly luciferase assay, 200 μl luciferase assay buffer (15 mM $K_3PO_4$ [pH 7.8], 25 mM glycylglycine, 15 mM $MgSO_4$, 4 mM EGTA) with freshly added 1 mM DTT, 2 mM ATP and 1 mM D-luciferin (PJK) was mixed with 20 μl luciferase lysis buffer and measured for 20 s. For renilla luciferase assay, 100 μl luciferase assay buffer supplemented with 1.43 μM coelenterazine (PJK) was mixed with 20 μl lysate and measured for 10 sec by using either a Lumat LB9507 tube luminometer or a Mithras LB940 plate luminometer (both from Berthold Technologies).

**Immunofluorescence microscopy**. Immunofluorescence microcopy was performed as described previously[57]. Cells cultured on glass coverslips were fixed with 4% paraformaldehyde in PBS for 30 min. The cells were permeabilized with PBS containing 0.1% Triton X-100, blocked with 5% FBS or BSA, and then incubated with diluted primary antibody for 60 min at room temperature. After washing with PBS three times, cells were incubated with Alexa-dye labeled secondary antibodies in PBS containing 5% FBS for 60 min. The coverslips were mounted in Fluoromount-G (SouthernBiotech) and images were obtained with a Leica SP8 confocal microscope.

**Image-based detection of SARS-CoV-2 infection and co-expressing cells**. Image-based quantification of SARS-CoV-2 infected cells was based on the immunofluorescence detection of nucleocapsid protein-positive cells. Cells were seeded into 96-well black wall imaging plates for 24 h, followed by infection with SARS-CoV-2 at an MOI of 0.5 in the case of Huh7-Lunet/T7-ACE2 cells or a MOI of 5 for Calu-3 cells. Thirty minutes post infection, inhibitors of PLD1/2 were added and cells were kept at 37 °C for 24 h. The cells were fixed with 4% paraformaldehyde, blocked with 1% skimmed milk, and incubated with anti-nucleocapsid antibody for 1 h at 4 °C, followed by counterstaining with donkey anti-mouse secondary antibody coupled to Alexa568. Nuclear DNA was stained with DAPI and cells were examined using a Nikon Ti2 spinning disk microscope equipped with a Plan Apo lambda 20x/0.75 air objective and a back-illuminated EM-CCD camera (Andor iXon DU-888). Segmentation of nuclei was done with the CellProfiler version 3.1.9 software package. Cytoplasm was identified by expanding the nuclei by 5 pixels. Separation of cells into infected and non-infected population was performed with a semi-supervised machine-learning-based approach using CellProfiler Analyst, as described earlier[58]. The schematic of the pipeline is outlined in Supplementary Fig. 9b. The enrichment score of cells co-expressing HA-nsp3 and GFP-PABD-Raf1 (in Fig. 8b), or expressing punctate pattern of AGPAT2 (Fig. 5b) and GFP-PABD-Raf1 (Fig. 7b), is calculated as described before[59]. In short, the enrichment score indicates the probability of the presence of a specific class (for example, co-expression or punctate expression pattern) in different samples in relation of the total cells in the samples. The scripts, training set and images used to train the classifier are available at https://doi.org/10.17632/b6hdc96ks5.1

**RT-qPCR assay for SARS-CoV-2 replication**. Total RNA was extracted from cells using NucleoSpin RNA extraction kit (Machery-Nagel) following the manufacturer's protocol. Reverse transcription (RT) reaction for cDNA synthesis was performed using the high-capacity cDNA RT kit (ThermoScientific). Each cDNA was diluted 1:5 in $H_2O$ and qPCR was performed using iTaq Universal SYBR green mastermix (Bio-Rad). Primers for qPCR were designed using Primer 3 for SARS-CoV-2-ORF1 (Forward 5′- GAGAGCCTTGTCCCTGGTTT-3′, Reverse 5′-AGTC TCCAAAGCCACGTACG-3′) and HPRT (Forward 5′-CCTGGCGTCGTGATT AGTG-3′, Reverse 5′-ACACCCTTTCCAAATCCTCAG-3′). Relative abundance for SARS-CoV-2 Orf1 mRNA was corrected for PCR efficiency and normalized to HPRT transcript level.

**SARS-CoV-2 DMV generation and quantification**. Based on earlier findings that coronavirus DMV formation can be induced by the sole expression of viral non-structural protein (nsp)3-4[12], we similarly constructed an expression vector containing the SARS-CoV-2 (nsp)3-4 genes tagged with HA (for nsp3) and V5 (for nsp4). Expression of this construct in Huh-derived cells showed significant number of DMVs in only the transfected cells, independent of infection. Using this system, we measured the number and diameter of DMVs and MMVs in Lunet cells under different perturnation modalities (Figs. 3f, 4e and f), as described before for HCV-induced DMV formation[22].

**Immunoprecipitation and immunoblotting**. For immunoprecipitation, cells were processed as described for proteome analysis, but captured protein complexes were eluted by using 2× sample buffer (100 mM Tris-HCl [pH 6.8], 4% SDS, 12% β-mercaptoethanol, 20% glycerol, 0.001% bromophenol blue). To prepare lysates for immunoblotting, cells were incubated in 2× lysis buffer (200 mM Tris [pH 8.8], 5 mM EDTA, 0.1% bromophenol blue, 10% sucrose, 3% SDS, 2% β-mercaptoethanol) and incubated for 5 min at 95 °C. Proteins were separated by SDS-polyacrylamide gel electrophoresis and electro-transferred onto PVDF membranes. After blocking of the membranes with 5% nonfat milk, they were incubated overnight at 4 °C with primary antibodies. After washing with 0.5% Tween 20 in PBS, membranes were incubated with secondary horseradish peroxidase-conjugated antibodies for 1 h at room temperature. Membranes were developed by

using Western Lightning Plus-ECL reagent (PerkinElmer), and signals were detected by Intas ChemoCam Imager 3.2 (Intas).

**Electron microscopy**. Transmission EM was performed as described previously[13]. In brief, cells were fixed with 2.5% glutaraldehyde (GA) in 50 mM sodium cacodylate buffer (CaCo), supplemented with 2% sucrose, 50 mM KCl, 2.6 mM $MgCl_2$ and 2.6 mM $CaCl_2$, for 30 min at room temperature. After five washes with 50 mM CaCo, samples were incubated with 2% $OsO_4$ in 25 mM CaCo for 40 min on ice, washed three times with EM-grade water and incubated in 0.5% uranyl acetate in water overnight at 4 °C. Samples were rinsed three times with EM-grade water, dehydrated in a graded ethanol series (from 40 to 100%) at room temperature, embedded in Epon 812 (Electron Microscopy Sciences) and polymerized for 2 days at 60 °C. After polymerization, ultrathin sections of 70 nm were obtained by sectioning with an ultramicrotome Leica EM UC6 (Leica Microsystems) and mounted on a slot grid. Sections were counterstained using 3% uranyl acetate in 70% methanol for 5 min and lead citrate (Reynold's) for 2 min and examined by using a JEOL JEM-1400 (JEOL) operating at 80 kV and equipped with a 4 K TemCam F416 (Tietz Video and Image Processing Systems GmbH).

**Correlative light and electron microscopy (CLEM)**. Two methods were employed using protocols described previously[22]. For CLEM analysis with low-resolution fluorescence imaging, $0.5 \times 10^5$ Huh7-Lunet/T7 cells were seeded onto glass-bottom culture dishes containing gridded coverslips (MatTek Corporation) and incubated overnight. Cells were transfected with plasmid pTM NS3-5B encoding the HCV replicase (NS3, NS4A, NS4B, NS5A and NS5B) with NS5A containing a GFP insertion that does not affect viral protein function[60]. For Fig. 3d, plasmid encoding AGPAT2-GFP was transfected into Huh7-Lunet/T7 cells. After 24 h, differential interference contrast (DIC) and GFP signals were acquired by using a widefield fluorescence microscope (Nikon Eclipse) with a 10x objective lens. Cells were fixed with 2.5% GA, 2% sucrose in 50 mM CaCo, supplemented with 50 mM KCl, 2.6 mM $MgCl_2$ and 2.6 mM $CaCl_2$ for 30 min at room temperature. After five washes with CaCo, cells were processed for EM analysis as described above.

For CLEM with high precision image correlation, cells were fixed with 4% paraformaldehyde and 0.2% GA in PBS for 30 min at room temperature, washed three times with 150 mM glycine in PBS, once with PBS, stained with LipidTox™ Deep Red Neutral Lipid Stain (Invitrogen) and analyzed by using a spinning disc confocal microscope. Fluorescence images were acquired with optical sections of 0.2 μm using a 100x objective. For low-precision CLEM (Figs. 3f and 4e), Huh7-Lunet/T7/Ctrl KO, Huh7-Lunet/T7/AGPAT2 KO or Huh7-Lunet/T7/DKO cells were seeded onto dishes containing gridded coverslips (MatTek Corporation). Twenty-four hours after transfection with pTM-SARS-CoV-2-HA-3-4-V5-2A-NG (NeonGreen) using TransIT-LT1 Transfection Reagent (Mirus Bio), samples were observed by confocal microscopy with a ×10 objective lens to locate NeonGreen-positive cells. After imaging, cells were fixed again with 2.5% GA, 2% sucrose in 50 mM CaCo, supplemented with 50 mM KCl, 2.6 mM $MgCl_2$ and 2.6 mM $CaCl_2$ for at least 30 min on ice. Further processing for EM analysis was the same as described above.

**Detection of PA in transiently permeabilized cells**. A recombinant protein composed of the PA-binding domain (BD) from yeast Spo20 fused to the N-terminus of glutathione-S-transferase (GST-PABD) was expressed in E. coli, strain BL21 (DE3), by using the expression vector pGEX-6P-1. Cells were grown in LB medium until OD600 = 0.5, followed by a 3 h-incubation period in medium containing 1 mM isopropyl β-d-1-thiogalactopyranoside (IPTG) at 37 °C. Cells were harvested by centrifugation and GST fusion proteins were purified from cell lysates by using a Spin Purification kit as recommended by the manufacturer (Thermo Fisher Scientific). Eluates were dialyzed in PBS using Slide-A-Lyzer Dialysis Cassettes (7 K molecular weight cut-off; Thermo Fisher Scientific) and purified proteins were stored in 50% glycerol in PBS at −20 °C at a concentration of 1 mg/ml. Detection of PA with the GST-PABD in semi-intact cells was performed as described previously[61], with slight modifications. In brief, cells seeded on coverslips were washed twice with PBS and incubated on ice with 200 ng/ml streptolysin O (SAE0089; Sigma Aldrich) in PBS for 5 min. After washing three times with PBS, cells were incubated with transport buffer (25 mM HEPES-KOH, [pH 7.4], 115 mM potassium acetate, 2.5 mM $MgCl_2$) at 37 °C for 5 min. After washing twice with transport buffer at room temperature, cells were incubated with a reaction mixture (1 mM ATP, 50 μg/ml creatine kinase, 2.62 mg/ml creatine phosphate, 1 mg/ml glucose, 1 mM GTP, and 5 μg/100 μl GST-PABD wild-type or 4E mutant) at 37 °C for 15 min. Cells were washed with transport buffer, fixed with 4% paraformaldehyde for 30 min, and permeabilized with 0.2% Triton X-100 for 20 min. After blocking with 5% skim milk, GST-PABD proteins were detected using a GST-specific mouse monoclonal antibody as primary antibody and an Alexa 488-conjugated anti-mouse antibody as a secondary antibody.

**Generation of VSVΔG-G and VSVΔG-S pseudovirus**. The generation of pseudovirus was adapted from Zettl and co-workers[62]. In brief, to generate replication-deficient G-protein decorated VSV particles (VSVΔG-G) encoding Firefly Luciferase and GFP, $5 \times 10^6$ BHK-G43 cells (kindly provided by Gert Zimmer) were seeded into a 10 cm-diameter dish and incubated overnight. The next day the culture medium was replaced by Glasgow's Minimal Essential Medium (GMEM)

(Thermo Fisher Scientific) with 5% FCS and 1 nM mifepristone (Sigma Aldrich) to induce VSV G protein expression. After 6 h the cells were infected with VSVΔG-G. After an overnight incubation the pseudovirus containing supernatant was filtered through a 0.45 μM filter, aliquoted and stored at −70 °C.

To produce propagation deficient VSVΔG pseudotyped with the SARS-CoV-2 Spike protein (VSVΔG-S), 9 × 10⁶ HEK-293T cells were seeded into a poly-L-lysine coated 10 cm-diameter dish. The next day the cells were transfected with an expression plasmid encoding a codon-optimized SARS-CoV-2 Spike protein lacking part of the C-terminal domain. Lipofectamine 2000 (Invitrogen) was used as transfection reagent. The cells were incubated overnight to allow the expression of the SARS-CoV-2 Spike protein. On the consecutive day cells were infected with VSVΔG-G for 2 h. Anti-VSV-G antibody-containing supernatant of I1 hybridoma cells (ATCC: CRL-2700™) was mixed at a ratio of 1:10 with culture media and added onto the cells. The cells were incubated for 30 min with the antibody, washed with DMEM cplt (see 'Cell culture and transfection' for details) and fresh culture medium was added onto the cells. After around 12 h incubation, culture supernatants containing VSVΔG-S pseudotyped viruses were harvested, filtered through a 0.45 μm filter and stored at −70 °C.

**Cell viability and cell growth assays**. Cell viability was measured by using the CellTiter-Glo luminescent cell viability assay (Promega) according to the protocol of the manufacturer. Cell growth was determined by cell counting using a TC20 automated cell counter (BioRad).

**Statistics and reproducibility**. Unless otherwise stated, values represent the mean of a given number of replicates. Error bars are SD or SEM as indicated in the figure legends. Student's *t*-tests were performed for unpaired or paired groups by using the GraphPad Prism 5.03 software package (GraphPad software). A *P* value < 0.05 was considered statistically significant. All experiments were repeated two or three times independently, as indicated in the figure legends. No statistical method was used to predetermine sample size. The experiments were not randomized and the investigators were not blinded to allocation during the experiments.

**Reporting summary**. Further information on research design is available in the Nature Research Reporting Summary linked to this article.

## Data availability

All data are available in the main text or the Supplementary Information. Source data are provided with this paper. The mass-spectrometry-based proteomics data generated in this study have been deposited to the ProteomeXchange Consortium via the PRIDE[63] partner repository with the dataset identifier PXD029692 with related raw data provided as supplementary dataset 1. The raw data for lipidome analysis is provided as supplementary dataset 3. Source data are provided with this paper.

## Code availability

The scripts, training set and sample images to train the classifier for Figs. 5b, 7b, and 8b are deposited at Mendeley Data (https://doi.org/10.17632/b6hdc96ks5.1).

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

## Acknowledgements

This work was supported by grants from the Deutsche Forschungsgemeinschaft (DFG, German Research Foundation) – Project Number 240245660 - SFB1129, Project Number 272983813 – TRR 179, Project Number 112927078 - TRR 83, Project Number 314905040 – TRR 209, and the Helmholtz Association's Initiative and Networking Fund (KA1-Co-02; "COVIPA"), all to R.B. B.B. was supported by the DFG, Project Number 112927078 - TRR 83, Project Number 240245660 - SFB1129, and Project Number 316659730. V.P. is supported by a European Molecular Biology Organization (EMBO) Long-Term Fellowship (ALTF 454-2020). C.J.N. was supported in part by a European Molecular Biology Organization (EMBO) Long-Term Fellowship (ALTF 466-2016). Y.S. is funded by the Hector Fellow Academy. L.K. and C.Z. acknowledge funding from the BMBF, grant number 031A602A (ERASysApp SysVirDrug). P.V. and V.T. were supported by the Swiss National Science foundation (SNF Project Number 310030_173085). G.S.-F. and K.H. were supported by a European Research Council Advanced Investigator Grant (ERC AdG 695214 i-FIVE). We thank Marie Bartenschlager, Lena Werstein, Ulrike Herian, Stephanie Kallis, Iris Leibrecht and Fredy Huschmand for excellent technical assistance. We are grateful to Eliana Acosta and Heeyoung Kim for editorial assistance. We acknowledge Alessia Ruggieri for providing plasmid constructs. We also acknowledge the excellent support provided by the Infectious Diseases Imaging Platform (IDIP) headed by Vibor Laketa, the University of Heidelberg Electron Microscopy Core Facility (EMCF Heidelberg) headed by Stefan Hillmer and the Proteomics and Metabolomics Facility (Pro-Met-) at CeMM. We thank the European Virus Archive (EVAg) and Christian Drosten for the provision of the SARS-CoV-2 isolate Bavpat1/2020 and EVAg for providing the HP/F/2013 ZIKV strain. We are grateful to Gerd Zimmer for providing the propagation deficient VSVΔG replicon and the BHK-643 cells that were used for the pseudovirus assays and Stephanie Pfänder at the University Bochum for initial help with setting up the VSV pseudotype assay. We also thank all members of the Molecular Virology unit for continuous stimulating discussions.

## Author contributions

Conceptualization, K.T., V.P., D.P., and R.B.; Investigation, K.T., V.P., D.P., J-Y.L., M.-T.P., W.-I.T., C.J.N., M.C., B.C., Y.S., S.J., C.-S.T., C.L., P.V., K.H., A.C.M, C.Z., U.H., J.B., L.K., B.B.; Writing – Original Draft, K.T., V.P., and R.B.; Writing – Review & Editing, K.T.,V.P., D.P., J.Y.L., C.J.N., A.M, A.C.M., L.K., H.E., V.T., G.S-F., B.B. and R.B.; Funding Acquisition, R.B., B.B., L.K. and G.S-F.

## Funding

## Competing interests

The authors declare no competing interests.
