## [Peer Review File · Nature Communications]

Reviewer comments, first round review -

Reviewer #1 (Remarks to the Author):

In this study the authors describe the role of two cellular acyltransferases AGPAT1/2 responsible for synthesis of PA that play a critical role in the biogenesis of replication organelles of 2 positive-sense RNA viruses - HCV and SARS-CoV-2. The mechanism of viral replication organelle biogenesis is a fundamental question in the lifecycle of RNA viruses which replicate in the cytosol. The authors show that synthesis of PA via AGPAT1/2 is critical for the formation of double membrane vesicles that morphologically resemble autophagosomes, underscoring the key function that lipids play in membrane reorganisation associated with formation of virus replication organelles. This is an important study that not only the flavivirus field but also generally +RNA virus biology will benefit from.

General comments:

Previous studies have shown that increase in cellular concentrations of PA and signaling via PA results in activation of mTOR, which should result in inhibition of the autophagy pathway (Fang et al, Science 2001). On the other hand infection by either HCV or by SARS-CoV-2 triggers activation of the autophagy pathway, and this study highlights the importance of PA synthesis in formation of DMVs and autophagosomes. It would therefore be useful if the authors could provide a discussion on what is the effect of HCV infection and PA synthesis on mTOR activation, and indeed how these results can be reconciled with each other.

Specific comments:

1. In Figure 1F, are the background bands (particularly detectable for AGPAT2) in the double knock-out cells non-specific? Or is this a pooled population of cells which might have residual wild-type cells?
2. In Figure 1G, the authors show very nicely that reconstitution of only the wt but not the catalytic mutants of the AGPATs can rescue the phenotype. It might also be useful to feed PA in the culture medium to test whether supplying the lipid itself to the KO cells will be able to rescue the defect in virus replication and DMV biogenesis observed in the SKO and DKO cells.
3. In Figure 2A, is the morphology of the DMVs altered in the KO cells? If so, would there be a way of quantifying the altered morphology (and not just the abundance) of the DMVs formed in the wt versus the SKO/DKO cells? In the same vein, is it possible to visualise what happens to autophagosomes in these HCV NS3-5B expressing cells themselves?
4. In Figure 2D, it's not clear what the EM image depicts? Also, in the quantification of the isolated DMVs, was PE not measured at all, or was it not detectable? The values presented are the lipid fractions associated with the replicon compared to control in wt cells. What happens to these values in the DKO cells?
It would also be useful to present these data as % of cellular lipid levels in replicon expressing cells compared to control cells – under physiological conditions, the cellular concentration of PA is ~5% of PC. It would be very useful to know the extent of alteration in these numbers in the infected wt

samples as well as in the KO samples.

5. In Figure 2E, did the authors express/test the PA sensor in the single and double knock-out cells to visualise cellular PA as a negative control?

6. Minor point in Fig 3C – the annotation says mock (indicative of infection), but really indicates empty vector.

7. In figure 4, is there any explanation for the differential effect of PLD1 and PLD2 inhibition?

Reviewer #2 (Remarks to the Author):

Tabata et al. use a system previously developed by the Bartenschlager laboratory to isolate HCV NS4B and associated membranes from cells in order to identify co-enriching cellular proteins. They identify 309 proteins that were significantly enriched in the sample isolated from NS4B-HA expressing cells compared to calnexin-HA expressing cells. Among these proteins are AGPAT1 and AGPAT2, which catalyze the synthesis of phosphatidic acid (PA). They found that AGPAT1/2 DKO cells had a significant inhibition of HCV replication but also of lipid droplet formation. In addition, DMV formation was significantly reduced by AGPAT knockout. PA was found to be more abundant on NS4B-associated membranes as well as at HCV-induced membrane alterations by microscopy.

The authors then asked whether AGPAT1/2 were relevant to SARS-CoV-2 replication. While AGPAT knockout cells showed reduced infection by SARS-CoV-2, there were no decreases in DMV formation using a nsp3-4 expression model. Inhibiting other pathways of PA production also decreased SARS-CoV-2 infection.

The work is generally well performed and the role of AGPATs in HCV replication as well as in SARS-CoV-2 infection is clearly demonstrated. The work is significant to the field by extending what is known about lipids in HCV replication/DMV formation and by drawing intriguing parallels between the role of PA in DMV formation in hepacivirus and in coronavirus infection.

Major concerns

- The protocol used for NS4B-HA purification in this work appears to have omitted the density gradient centrifugation step used in the earlier Paul et al. manuscript. If so, it is misleading to state that “pull-down of NS4B-associated membrane fractions was performed as described previously” and this statement should be corrected. This raises concern that the final “purified” membrane fraction is not equivalent to the membranes that were characterized in detail in the original Paul et al. manuscript. While this does not materially affect the AGPAT data, it does raise questions about the significance of the other proteins identified on mass spectrometry as well as the lipidome analysis. Additional experiments should be performed to demonstrate the degree of purification by the single-step purification assay and compare it to the two-step assay. A total protein stain of the purified proteins compared to the CANX-HA sample would be useful in this regard, for example.
- A decrease in SARS-CoV-2 infection by AGPAT knockout could be due, of course, to inhibition of entry and viral assembly/secretion, not just replication. The experiment shown in Fig 3E is a

multicycle infection. This is particularly relevant because AGPAT knockout did not appear to affect the number of DMVs induced by SARS-CoV-2 nsp3-4 expression. The authors should either modify their interpretation of their data to discuss infection (and not replication), or show assays directed specifically at SARS-CoV-2 replication. Do AGPATs have a role in SARS-CoV-2 entry?

- Machine learning algorithms are used for classification of various types of imaging data. As these are relatively new methods that can have significant variability in performance (e.g. due to quality of training sets, classifiers, etc) there should be some data to show the validity of these classifications and some additional information in the methods. For example, how many images were used for the training sets? How did the final automated classifier perform compared to manual classification by a blinded human? Notably, the citation provided in the manuscript (for adenovirus infection) surely did not use the same classification model used for this study. In addition, for the purposes of scientific reproducibility, the complete CellProfiler Analyst files used to perform these analysis (e.g. classification models, pipelines, training sets, etc) must be uploaded to a publicly accessible server or repository.

Minor concerns

- In panel 3E-G, only AGPAT2 KO is shown for the “SKO” data. What about AGPAT1?
- Another possible role of PA at HCV DMVs is for exchange with PI, e.g. by transfer proteins such as Nir2, which has been shown to have a role in HCV replication; this should be cited.
- The figure legend for panel 3(G) lacks the “(G)”.

Reviewer #3 (Remarks to the Author):

In this paper, the authors investigated the role of phosphatidic acid in the formation of replication organelles in the context of cell infected by HCV or SARS-CoV-2. This project was initiated on HCV by using a proteomic approach to determine the molecular composition of double-membrane vesicles (DMV) induced by HCV replication. Among the proteins identified, they selected a series of candidates based on their potential involvement in RNA metabolism, vesicle biogenesis and transport and membrane organization. Some of these factors were validated by siRNA screening. Amongst identified hits, AGPAT1 and AGPAT2 were selected for further investigation. By using CLEM technique, they showed that these proteins are recruited during the biogenesis of DMV induced by HCV replication. Their knock out affected HCV replication and DMV formation. The reaction product of AGPAT1 and AGPAT2 is phosphatidic acid and they also investigated the effect of HCV replication on the subcellular localization of this lipid. Phosphatidic acid was shown to accumulate in DMV by analyzing their lipidomic after purification and by using a phosphatidic acid sensor to demonstrate its accumulation in DMV in infected cells or cells expression HCV proteins responsible for the production of these vesicles. They also used pharmacological inhibitors to show that alternative pathways involved in the generation of phosphatidic acid can also play a role in HCV replication. Interestingly, the authors showed that AGPATs are also involved in SARS-CoV-2 and in the formation of autophagosome-like structures. Although the effect on SARS-CoV-2 was less dramatic than what they observed for HCV. These observations are novel and very interesting, and the experimental work is solid.

Specific points:

1-The authors only used pharmacological inhibitors to determine whether alternative pathways involved in the generation of phosphatidic acid can also play a role in HCV replication. These data

would be stronger if they could also validate by siRNA or CRISPR/Cas9.

2-Abstract line 60: it is inaccurate to state that phosphatidic acid is an important lipid used for replication organelle formation by HCV and SARS-CoV-2. Data presented in figures 3 and 4 indicate a change of the morphology of the DMV induced by SARS-CoV-2 protein expression, rather than a role in the formation of these structures.

3-line 358: the authors need to cite the paper by Wang and Tai (PMID: 31484747) in support to the hypothesis of a role of phosphatidic acid in an exchange lipid in a counter-transporter chain.

4-Figure 3G: there is no legend.

REVIEWER COMMENTS

Reviewer #1 (Remarks to the Author):

In this study the authors describe the role of two cellular acyltransferases AGPAT1/2 responsible for synthesis of PA that play a critical role in the biogenesis of replication organelles of 2 positive-sense RNA viruses - HCV and SARS-CoV-2. The mechanism of viral replication organelle biogenesis is a fundamental question in the lifecycle of RNA viruses which replicate in the cytosol. The authors show that synthesis of PA via AGPAT1/2 is critical for the formation of double membrane vesicles that morphologically resemble autophagosomes, underscoring the key function that lipids play in membrane reorganisation associated with formation of virus replication organelles. This is an important study that not only the flavivirus field but also generally +RNA virus biology will benefit from.

General comments:

Previous studies have shown that increase in cellular concentrations of PA and signaling via PA results in activation of mTOR, which should result in inhibition of the autophagy pathway (Fang et al, Science 2001). On the other hand infection by either HCV or by SARS-CoV-2 triggers activation of the autophagy pathway, and this study highlights the importance of PA synthesis in formation of DMVs and autophagosomes. It would therefore be useful if the authors could provide a discussion on what is the effect of HCV infection and PA synthesis on mTOR activation, and indeed how these results can be reconciled with each other.

We appreciate this suggestion to include a discussion on the effect of HCV infection and PA synthesis on mTOR activation. However, we would like to point out that although HCV activates autophagy, it is not required per se for viral replication, but rather distinct components of the autophagy machinery (Mori et al., Journal of General Virology 2018;99:1643-1657). Moreover, the effect of HCV infection on mTOR activation is discussed controversially. For instance, Shrivastava S. et al. reported that HCV induces autophagy and activates mTOR signaling pathway as determined by monitoring phospho-4E-BP1¹. In contrast, Huang H. et al. showed that mTOR activity as monitored by, amongst others, phospho-p70 S6-kinase is inhibited by HCV². Nevertheless, we investigated the effect of PA synthesis on mTOR activation during

HCV infection (Figure R1). Using Huh7.5 cells infected with HCV Jc1 (MOI=1) in the presence or absence of general AGPAT and PLD2 inhibition, we determined whether mTOR activation is differentially regulated by measuring the phosphorylation of the p70 S6-kinase subunit and 4E-BP1. Since basal mTOR may be activated in the presence of serum, we also included cells that were serum-starved. Even under these conditions, we could not detect significant mTOR activation over the basal levels during HCV infection. However, lower relative levels of phospho-p70 S6-kinase and 4E-BP1 were observed in PA inhibitor treated cells, confirming the positive role of PA in mTOR activation. It is unclear why PA dependent mTOR activation would inhibit autophagy and yet promote HCV replication. A point to consider, as mentioned above, is the fact that distinct components of the autophagy machinery, but not autophagy per se, are required for the replication of HCV and SARS-CoV-2. In addition, since we showed the local generation of PA at viral replication sites via the recruitment of AGPAT proteins, mTOR might be activated only locally at these sites, but not at the whole cell/cell population level. It should also be noted that even though PA positively regulates mTOR activation, it plays a positive role in bulk autophagy³. For these reasons, we included a short paragraph to the reviewer's point in the manuscript, but did not include the data in Figure R1.

Figure R1: Huh7.5 cells were inoculated with HCV Jc1 (MOI= 1). At 4 h.post infection cells were further cultured in either 10% FCS or FCS-free medium containing AGPAT (CI976) or PLD2 (ML298) inhibitors as indicated. At 48 h.post infection cells were harvested and

subjected to immunoblotting with antibodies detecting total or phospho-specific p70 S6 and 4E-BP1 to monitor mTOR pathway activation.

Specific comments:

1. In Figure 1F, are the background bands (particularly detectable for AGPAT2) in the double knock-out cells non-specific? Or is this a pooled population of cells which might have residual wild-type cells?

This is indeed true, we used cell pools to avoid single cell clone-specific effects. Huh7.5 cells were infected with lentiviruses encoding AGPAT-specific sgRNA and selected with puromycin for 5 days. Thus, the most likely explanation is that in some cells in the pool, AGPAT2 is still expressed, giving rise to the residual signal in western blotting when using bulk measurements. We specified in the manuscript the use of cell pools.

2. In Figure 1G, the authors show very nicely that reconstitution of only the wt but not the catalytic mutants of the AGPATs can rescue the phenotype. It might also be useful to feed PA in the culture medium to test whether supplying the lipid itself to the KO cells will be able to rescue the defect in virus replication and DMV biogenesis observed in the SKO and DKO cells.

Indeed, we tried such a rescue experiment by feeding commercially available 16:0-18:0 PA (Avanti, 840857) to AGPAT KO cells. These lipids had to be dissolved in chloroform, because solubility in DMSO or Ethanol is insufficient. The PA/Chloroform mixture was added to medium of AGPAT DKO cells (Figure R2). Using concentrations between 7.8125 or 31.25 μ M did not affect cell viability and had no impact on HCV replication, while higher concentrations caused cell lysis. However, we noted that the PA/Chloroform organic phase separated from the medium and therefore, the actual lipid concentration was most likely much lower, but the actual concentration could not be determined. We added a brief comment to that in the main text.

PA concentration: (μM): 0,00 0,12 0,49 1,95 7,81 31,25 125,00 500,00 2000,00

Figure R2: Control or AGPAT DKO Huh7-Lunet cells were electroporated with subgenomic HCV replicon transcripts of (sgJFH) and were with PA dissolved in Chloroform at a concentration of 2000, 500, 125, 31.25, 7.8, 1.95, 0.49, 0.12 or 0 μM . Cells were lysed at 24-, 48- or 72-hours post-electroporation. HCV replication and cell viability were measured by *firefly* luciferase or CellTiter-Glo (Promega) assay, respectively. In the top left panels, raw values of luciferase counts are shown as RLU for each time point. In the top right panels, CellTiter-Glo counts from samples treated with 0 μM PA/Chloroform

treatment at higher concentrations severely affected cell viability, which was much less the case at lower concentrations. Therefore, in the bottom graph, values obtained at lower concentrations were extracted from top panels and are shown as bar graph.

3. In Figure 2A, is the morphology of the DMVs altered in the KO cells? If so, would there be a way of quantifying the altered morphology (and not just the abundance) of the DMVs formed in the wt versus the SKO/DKO cells? In the same vein, is it possible to visualise what happens to autophagosomes in these HCV NS3-5B expressing cells themselves?

Indeed, in Figure S4C we had already provided analysis of DMV diameter that was reduced in AGPAT KO cells. Further inspection of microscopic images for DMVs and double membrane tubules induced by HCV NS3-5B expression did not reveal obvious morphological alterations in AGPAT SKO or DKO cells compared to wild-type cells.

4. In Figure 2D, it's not clear what the EM image depicts? Also, in the quantification of the isolated DMVs, was PE not measured at all, or was it not detectable? The values presented are the lipid fractions associated with the replicon compared to control in wt cells. What happens to these values in the DKO cells?

It would also be useful to present these data as % of cellular lipid levels in replicon expressing cells compared to control cells – under physiological conditions, the cellular concentration of PA is ~5% of PC. It would be very useful to know the extent of alteration in these numbers in the infected wt samples as well as in the KO samples.

We apologize for the misunderstanding. What is shown are example EM images of purified DMVs and ER membranes. Moreover, lipidome analysis was not done on whole cells but rather on purified DMVs and ER membranes that were used for normalization. To make these points clear we have modified the figure legend that reads:

“(D) Lipidome analysis of HCV induced DMVs. Extracts of Huh7 cells containing the subgenomic replicon sg4BHA31R (NS4B-HA) and Huh7 cells stably overexpressing HA-tagged Calnexin (CNX-HA) and control Huh7 cells were prepared as described in supplementary methods and used for HA affinity purification under native conditions to enrich for DMVs (NS4B-HA) and for ER membranes that served as reference (CNX-HA). An aliquot of the sample was analyzed by electron microscopy (top panels)

whereas the majority was subjected to lipidome analysis by using mass spectrometry. Representative membrane structures are shown on the top: DMV-like vesicles in the NS4B-HA sample (left) and single membrane tubes in the CNX-HA sample (right). Amounts of selected lipids determined by MS for the NS4B-HA sample were normalized to those obtained for the CNX-HA sample that was set to one (bottom panel). The complete list of analyzed lipids is summarized in data S3.”

5. In Figure 2E, did the authors express/test the PA sensor in the single and double knock-out cells to visualise cellular PA as a negative control?

Thank you for this suggestion. We have included these new data into supplementary Figure S6 to show subcellular distribution of the PA sensor in wild-type, single (SKO) and double knock-out (DKO) cells in the presence of HCV-NS3-5. As expected, we observed diffuse cytoplasmic distribution of the sensor in SKO and DKO cells.

6. Minor point in Fig 3C – the annotation says mock (indicative of infection), but really indicates empty vector.

We have rectified this error.

7. In figure 4, is there any explanation for the differential effect of PLD1 and PLD2 inhibition?

There may be complementarity in the activity of PLD1 and PLD2 for PA production (as shown in Fig 2F). Another possibility is the complementation of PA synthesis by other PA enzymes. We have added a brief comment in the manuscript

Reviewer #2 (Remarks to the Author):

Tabata et al. use a system previously developed by the Bartenschlager laboratory to isolate HCV NS4B and associated membranes from cells in order to identify co-enriching cellular proteins. They identify 309 proteins that were significantly enriched in the sample isolated from NS4B-HA expressing cells compared to

calnexin-HA expressing cells. Among these proteins are AGPAT1 and AGPAT2, which catalyze the synthesis of phosphatidic acid (PA). They found that AGPAT1/2 DKO cells had a significant inhibition of HCV replication but also of lipid droplet formation. In addition, DMV formation was significantly reduced by AGPAT knockout. PA was found to be more abundant on NS4B-associated membranes as well as at HCV-induced membrane alterations by microscopy.

The authors then asked whether AGPAT1/2 were relevant to SARS-CoV-2 replication. While AGPAT knockout cells showed reduced infection by SARS-CoV-2, there were no decreases in DMV formation using a nsp3-4 expression model. Inhibiting other pathways of PA production also decreased SARS-CoV-2 infection.

The work is generally well performed and the role of AGPATs in HCV replication as well as in SARS-CoV-2 infection is clearly demonstrated. The work is significant to the field by extending what is known about lipids in HCV replication/DMV formation and by drawing intriguing parallels between the role of PA in DMV formation in hepacivirus and in coronavirus infection.

Major concerns

- The protocol used for NS4B-HA purification in this work appears to have omitted the density gradient centrifugation step used in the earlier Paul et al. manuscript. If so, it is misleading to state that “pull-down of NS4B-associated membrane fractions was performed as described previously” and this statement should be corrected.

We apologize for this mistake, as indeed, we omitted the density gradient centrifugation, which we did to increase yields. This purification protocol used for proteomic and lipidomic analysis evolved from the earlier study and was instrumental to increase the yield, which became possible by using the MACS system instead of “regular HA beads”. Given the fact that the MACS column was incompatible with high density sucrose because of viscosity, we directly applied the PNS (omitting the density gradient step). This has been specified in the materials and methods section of the revised manuscript and in the legend to supplementary Figure S1.

This raises concern that the final “purified” membrane fraction is not equivalent to the membranes that were characterized in detail in the original Paul et al. manuscript. While this does not materially affect the AGPAT data, it does raise questions about the significance of the other proteins identified on mass spectrometry as well as the lipidome analysis. Additional experiments should be performed to demonstrate the degree of purification by the single-step purification assay and compare it to the two-step assay. A total protein stain of the purified proteins compared to the CANX-HA sample would be useful in this regard, for example.

As stated above, the protocol had been changed mostly to increase the yield. We also want to point out that significance of hit candidates arises from the comparison with the control, which was purified in the same way. Finally, hit candidates were subject to validation by siRNA screening. Nevertheless, to address the reviewer’s comment, we have included additional data in supplementary Figure S1 of the revised manuscript that support our statement.

- A decrease in SARS-CoV-2 infection by AGPAT knockout could be due, of course, to inhibition of entry and viral assembly/secretion, not just replication. The experiment shown in Fig 3E is a multicycle infection. This is particularly relevant because AGPAT knockout did not appear to affect the number of DMVs induced by SARS-CoV-2 nsp3-4 expression. The authors should either modify their interpretation of their data to discuss infection (and not replication), or show assays directed specifically at SARS-CoV-2 replication. Do AGPATs have a role in SARS-CoV-2 entry?

A role of AGPATs in SARS-CoV-2 entry is indeed a possibility although some of our data already suggest that the primary role of PA is in HCV and SARS-CoV-2 replication. For example, a significant reduction in the intracellular SARS-CoV-2 viral RNA was observed in the presence or absence of PA inhibitors at 8 h post infection (Fig S10B). This time point most likely reflects a single round of infection and hence the effect of PA inhibition on viral spread (assembly and egress of virus particles and a new round of infection, i.e. viral entry) is most likely not measured. In addition, the drugs targeting PA enzymes were added 2 h after inoculation and hence their effect on viral entry are not reflected in these measurements.

Nevertheless, we followed the reviewer’s suggestion and conducted SARS-CoV-2 entry assays by using pseudotypes (VSVΔG-S pseudotyped with modified SARS-CoV-2

spike proteins). We show that AGPAT1/2 single and double KOs do not affect entry. This data is further supported by measuring the efficiency of VSVΔG-S entry in the presence of the general AGPAT inhibitor, CI976 that was added before or after pseudotype addition. These data have been included in Fig S9C of the revised manuscript and experimental approach is given in the material and methods section. Obtained results support our conclusion that AGPATs primarily play a role in SARS-CoV-2 replication, but not in the entry process.

- Machine learning algorithms are used for classification of various types of imaging data. As these are relatively new methods that can have significant variability in performance (e.g. due to quality of training sets, classifiers, etc) there should be some data to show the validity of these classifications and some additional information in the methods. For example, how many images were used for the training sets? How did the final automated classifier perform compared to manual classification by a blinded human? Notably, the citation provided in the manuscript (for adenovirus infection) surely did not use the same classification model used for this study. In addition, for the purposes of scientific reproducibility, the complete CellProfiler Analyst files used to perform these analysis (e.g. classification models, pipelines, training sets, etc) must be uploaded to a publicly accessible server or repository.

To demonstrate the validity of the machine learning algorithm used in our measurements, we have included the confusion matrix for various classifiers that will become available at Mendeley Data as indicated in supplementary information. These matrices compare the accuracy of prediction of the trained classifier for scoring classes relative to the correct classes scored by a blinded user. As a rule of thumb, the accuracy of prediction for the classes plotted in Fig 3B, 4B and 4D were between 90-95%.

The classifier model used in our machine learning classification was Random Forest classifier. The same classification model was used for the adenovirus infection associated cell cycle phase determination in the classifier employed in the cited manuscript.

Based on the suggestion of the reviewer, we have uploaded the CellProfiler scripts, training set, classifier details, confusion matrices, and the images used to train the Classifier for Fig 3B, 4B and 4D. These data are uploaded at Mendeley Data and the DOI are reserved (<http://dx.doi.org/10.17632/b6hdc96ks5.1>). The data will be made public upon acceptance of the paper. A copy of the uploaded data is available at

<https://heibox.uni-heidelberg.de/d/8e7722c293634f7ca16e/> with password - pamanuscript

Minor concerns

- In panel 3E-G, only AGPAT2 KO is shown for the “SKO” data. What about AGPAT1?

Most of our SARS-CoV-2 analyses focused on AGPAT2 KO cells that were used as such or for additional “transient” knockout of AGPAT1 (Fig3E), because stable DKO had long-term effects on cell viability (Fig. S3B). For the sake of uniformity and simplicity, we stayed in the same format. Also, since the effect of AGPAT1 and AGPAT2 single knockout on HCV replication was comparable, we did not include separate AGPAT1 knockout cells for EM experiments and rather used the general AGPAT inhibitor, CI976, for measuring the effect on SARS-CoV-2 DMVs (Fig 4F).

- Another possible role of PA at HCV DMVs is for exchange with PI, e.g. by transfer proteins such as Nir2, which has been shown to have a role in HCV replication; this should be cited.

This possibility is discussed in the manuscript, along with a recent reference where a similar suggestion has been made (Wang and Tai, J Virol 2019 Oct 29;93(22):e00742-19).

- The figure legend for panel 3(G) lacks the “(G)”.

Thank you, we have rectified this error.

Reviewer #3 (Remarks to the Author):

In this paper, the authors investigated the role of phosphatidic acid in the formation of replication organelles in the context of cell infected by HCV or SARS-CoV-2. This project was initiated on HCV by using a proteomic approach to determine the molecular composition of double-membrane vesicles (DMV) induced by HCV replication. Among the proteins identified, they selected a series of candidates based on their potential

involvement in RNA metabolism, vesicle biogenesis and transport and membrane organization. Some of these factors were validated by siRNA screening. Amongst identified hits, AGPAT1 and AGPAT2 were selected for further investigation. By using CLEM technique, they showed that these proteins are recruited during the biogenesis of DMV induced by HCV replication. Their knock out affected HCV replication and DMV formation. The reaction product of AGPAT1 and AGPAT2 is phosphatidic acid and they also investigated the effect of HCV replication on the subcellular localization of this lipid. Phosphatidic acid was shown to accumulate in DMV by analyzing their lipidomic after purification and by using a phosphatidic acid sensor to demonstrate its accumulation in DMV in infected cells or cells expression HCV proteins responsible for the production of these vesicles. They also used pharmacological inhibitors to show that alternative pathways involved in the generation of phosphatidic acid can also play a role in HCV replication. Interestingly, the authors showed that AGPATs are also involved in SARS-CoV-2 and in the formation of autophagosome-like structures. Although the effect on SARS-CoV-2 was less dramatic than what they observed for HCV. These observations are novel and very interesting, and the experimental work is solid.

Specific points:

1-The authors only used pharmacological inhibitors to determine whether alternative pathways involved in the generation of phosphatidic acid can also play a role in HCV replication. These data would be stronger if they could also validate by siRNA or CRISPR/Cas9.

Thank you for this suggestion. To support the pharmacological inhibition data of PLD1 and PLD2, we have added the suggested siRNA knockdown experiments that we performed in Huh7.5 cells infected with the HCV JC1 strain. We observed a consistent and significant reduction of HCV infection using cells depleted for PLD1 and PLD2. These new data support our conclusion and have been added in Fig S6c of the revised manuscript.

2-Abstract line 60: it is inaccurate to state that phosphatidic acid is an important lipid used for replication organelle formation by HCV and SARS-CoV-2. Data presented in figures 3 and 4 indicate a change of the morphology of the DMV induced by SARS-CoV-2 protein expression, rather than a role in the formation of these structures.

We agree to this view and modified the text accordingly.

3-line 358: the authors need to cite the paper by Wang and Tai (PMID: 31484747) in support to the hypothesis of a role of phosphatidic acid in an exchange lipid in a counter-transporter chain.

Thank you for this suggestion, we have added the reference.

4-Figure 3G: there is no legend.

Thank you, we have rectified this error.

References

1. Shrivastava S, Bhanja Chowdhury J, Steele R, Ray R, Ray RB. Hepatitis C virus upregulates Beclin1 for induction of autophagy and activates mTOR signaling. *J Virol* **86**, 8705-8712 (2012).
2. Huang H, Kang R, Wang J, Luo G, Yang W, Zhao Z. Hepatitis C virus inhibits AKT-tuberous sclerosis complex (TSC), the mechanistic target of rapamycin (MTOR) pathway, through endoplasmic reticulum stress to induce autophagy. *Autophagy* **9**, 175-195 (2013).
3. Holland P, *et al.* HS1BP3 negatively regulates autophagy by modulation of phosphatidic acid levels. *Nat Commun* **7**, 13889 (2016).

Reviewer comments, second round review -

Reviewer #1 (Remarks to the Author):

I am satisfied with the revised version of the manuscript.

Reviewer #2 (Remarks to the Author):

The authors have responded to my comments and questions; this work will be of significance to the field. I have no other concerns.

Reviewer #3 (Remarks to the Author):

I am satisfied with the modified version of the manuscript.